# Crystal structure and functional implications of cyclic di-pyrimidine-synthesizing cGAS/DncV-like nucleotidyltransferases

**Chia-Shin Yang[1,7], Tzu-Ping Ko [2,7], Chao-Jung Chen [3,4], Mei-Hui Hou[1], Yu-Chuan Wang [5] & Yeh Chen [6]** ✉

Purine-containing nucleotide second messengers regulate diverse cellular activities. Cyclic di-pyrimidines mediate anti-phage functions in bacteria; however, the synthesis mechanism remains elusive. Here, we determine the high-resolution structures of cyclic di-pyrimidine-synthesizing cGAS/DncV-like nucleotidyltransferases (CD-NTases) in clade E (CdnE) in its apo, substrate-, and intermediate-bound states. A conserved (R/Q)xW motif controlling the pyrimidine specificity of donor nucleotide is identified. Mutation of Trp or Arg from the (R/Q)xW motif to Ala rewires its specificity to purine nucleotides, producing mixed purine-pyrimidine cyclic dinucleotides (CDNs). Preferential binding of uracil over cytosine bases explains the product specificity of cyclic di-pyrimidine-synthesizing CdnE to cyclic di-UMP (cUU). Based on the intermediate-bound structures, a synthetic pathway for cUU containing a unique 2'3'-phosphodiester linkage through intermediate pppU[3'−5']pU is deduced. Our results provide a framework for pyrimidine selection and establish the importance of conserved residues at the C-terminal loop for the specificity determination of CD-NTases.

Cyclic dinucleotides (CDNs) are secondary messengers that regulate various cellular activities[1]. In bacteria, cyclic diguanylate (c-di-GMP, cGG) is synthesized by diguanylate cyclases with a GGDEF motif and binds to effectors, including PilZ-domain proteins and riboswitches that regulate important bacterial activities such as biofilm formation[2]. Cyclic diadenylate (c-di-AMP, cAA) is synthesized by proteins containing the diadenylate cyclase domain, and participates in cellular processes by binding to effector proteins and riboswitches[3]. In mammals, 2',3'-cyclic GMP-AMP (2',3'-cGAMP) is synthesized by cGAS, a cyclic GMP-AMP synthase activated by double-strand DNA[4]. The binding of 2',3'-cGAMP to stimulator of interferon genes (STING) effector proteins can induce innate immune responses in mammalian cells[5]. STING proteins also bind to cGG, cAA and 3',3'-cyclic GMP-AMP (cGA) and may have a bacterial origin[6,7]. The cGAS homolog VcDncV

(dinucleotide cyclase in *Vibrio cholerae*) synthesizes 3',3'-cGAMP, which binds to and activates VcCapV (CDN-activated phospholipase in *V. cholerae*), which degrades the bacterial cell membrane[8,9]. The DncV-CapV pair constitutes an example of a cyclic oligonucleotide-based anti-phage signaling system (CBASS), which protects the bacterial population from viral destruction through an abortive infection mechanism by inducing cell death[10,11].

Recently, scores of cGAS/DncV-like nucleotidyltransferases (CD-NTases) have been widely identified in bacterial genomes. CD-NTases can be classified into eight clades from A to H based on sequence identity, which usually corresponds to their ecological niches[12]. CD-NTases produce a variety of CDNs and cyclic trinucleotides (CTNs), including 3',3'-cyclic UMP-AMP (cUA) synthesized by *Rm*CdnE from *Rhodothermus marinus*, 3',3'-cyclic di-UMP (cUU) synthesized by

[1]Genomics BioSci & Tech Co. Ltd., New Taipei 221, Taiwan. [2]Institute of Biological Chemistry, Academia Sinica, Taipei 115, Taiwan. [3]Graduate Institute of Integrated Medicine, China Medical University, Taichung 406, Taiwan. [4]Proteomics Core Laboratory, Department of Medical Research, China Medical University Hospital, Taichung 404, Taiwan. [5]Trade Wind Biotech Co. Ltd., Taipei 115, Taiwan. [6]Department of Food Science and Biotechnology, National Chung Hsing University, Taichung 402, Taiwan. [7]These authors contributed equally: Chia-Shin Yang, Tzu-Ping Ko. ✉e-mail: chyeah6599@nchu.edu.tw

*Lp*CdnE from *Legionella pneumophila*, 3′,3′-cyclic UMP-GMP (cUG) synthesized by *Bd*CdnG from *Bradyrhizobium diazoefficiens*, 3′,2′-cGAMP synthesized by *As*CdnG from *Asticcacaulis* sp., 3′,3′,3′-cyclic AMP-AMP-AMP (cAAA) synthesized by *Ec*CdnC from *Escherichia coli* and 3′,3′,3′-cyclic AMP-AMP-GMP (cAAG) synthesized by *Ec*CdnD from

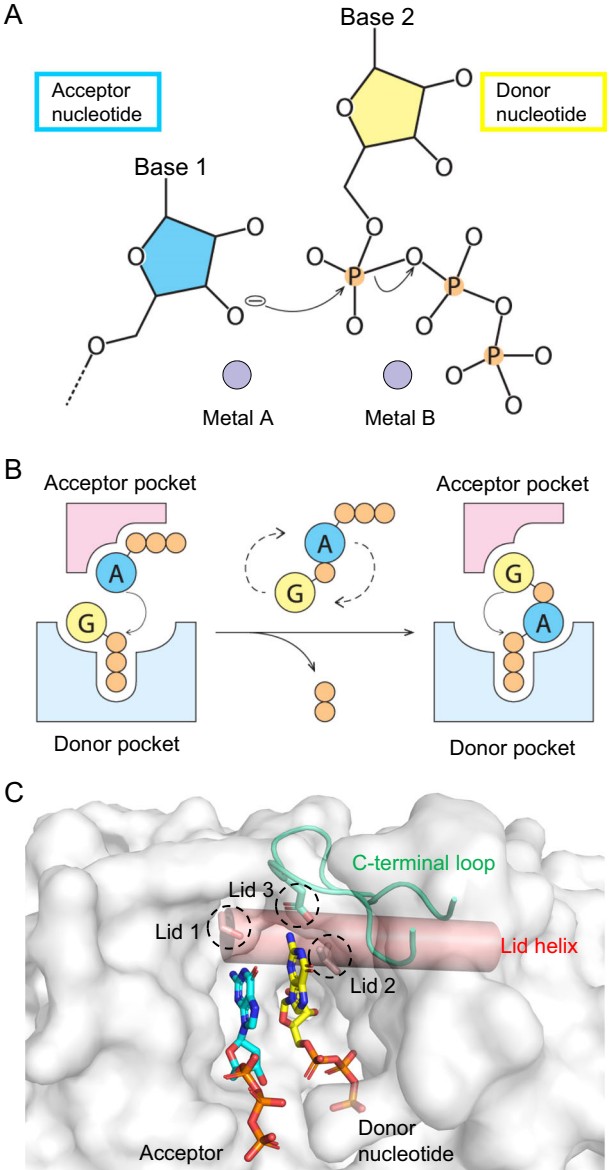

**Fig. 1 | Substrate recognition and catalytic pathway of CD-NTases. A** In the conserved canonical catalytic mechanism of nucleotidyltransferases, the ribose O3′ (or O2′) is deprotonated and stabilized by divalent cation "Metal A" before its attack at the P atom of the α-phosphate. The negative charge developed on the α-phosphate is neutralized by divalent cation "Metal B", which also binds to the β and γ-phosphate and leaves with the diphosphate group. Metal A and Metal B could be either $Mg^{2+}$ or $Mn^{2+}$. **B** In CD-NTases, the cage-like architecture helps retaining and reorienting the intermediate within the active site. In this example of *Vc*DncV with ATP as the acceptor and GTP as the donor, the ATP moiety of the intermediate pppApG takes over the original position of GTP, and vice versa, after the first reaction. The guanosine then serves as the acceptor and the ATP part as the donor in the second reaction. **C** Surface presentation of *Vc*DncV (PDB: 4XJ3). The lid helix (salmon) and the C-terminal loop (green) above the ligand-binding pocket are highlighted and indicated. The donor (yellow) and acceptor (cyan) nucleotide and the two conserved residues in lid helix, lid residue 1 (Lid 1) and lid residue 2 (Lid 2), are shown in sticks. In this study, we identified conserved lid residue 3 (Lid 3) located at the C-terminal loop, which dictates the donor nucleotide specificity.

*Enterobacter cloacae*[11–14]. The diversity of CDNs and CTNs is a consequence of the arms race between phage-bacteria and bacteria-host under evolutionary pressure, which could strengthen the anti-phage ability of bacteria[15] or facilitate evasion from innate immune detection by mammalian host cells[12].

CD-NTases share an N-terminal core domain with templated and non-templated polymerases of the nucleotidyltransferase (NTase) superfamily, but the C-terminal helical domain shows a higher degree of variation[12,16]. Three conserved acidic residues in the central β-sheet of the NTase domain coordinate with two metal ions (A and B) that are essential for catalysis[16]. The reaction proceeds by an attack of the acceptor nucleotide, whose O3′ (or O2′) atom is bound to metal A, at the α-phosphate of the donor nucleotide, whose triphosphate associates with metal B (Fig. 1A)[17]. The cage-like architecture of CD-NTase may allow retention and re-orientation of the intermediate molecule for the next step of catalysis[13,16]. Crystal structures of *Vc*DncV showed that the reaction intermediate had the original acceptor nucleotide bound to the donor site, and vice versa, while the second reaction took place, turning out the CDN product (Fig. 1B)[18]. Instead of base pairing, substrate specificity is determined by interactions with amino acid side chains surrounding the enzyme's active site[16]. Govande et al.[19] proposed that two key residues, lid residue 1 (lid 1) above the acceptor-nucleotide binding site and lid residue 2 (lid 2) above the donor pocket, were highly correlated with the product specificity of CD-NTases (Fig. 1C). For example, a conserved asparagine residue (lid 1) of CD-NTase in clade E (CdnE) has been shown to form two hydrogen bonds with the uracil base of the acceptor UTP in the structure of *Rm*CdnE[12]. Asparagine-to-serine reprogramming results in the pyrimidine-purine specificity alteration of *Rm*CdnE$^{N166S}$, which produces mainly cAA instead of cUA[12]. However, the currently available structures of CD-NTase show that lid 2 only makes non-specific interactions with the base or ribose of donor nucleotides, indicating that the proposed lid 2 is insufficient to explain the substrate specificity.

In contrast to the well-known biological roles of cyclic di-purines, including cGG, cGA, and cAA, the functional significance of cyclic di-pyrimidines has only recently begun to be appreciated. In this work, we aimed to determine the structural and functional character of cyclic di-pyrimidine-synthesizing CdnE, including *Lp*CdnE, *Cl*CdnE from *Cecembia lonarensis* and *Ef*CdnE from *Enterococcus faecalis*, which are encoded in type I CBASS operons with a transmembrane (TM) effector, named Cap15[20] (Fig. 2A). All three CdnEs contain Asn as lid 1 but *Cl*CdnE and *Ef*CdnE harbors His as lid 2, which differs from the Asn as lid 2 of *Lp*CdnE[19]. Structural characterization of these three CdnEs confirms the uracil-recognition role of the lid1 (Asn) for acceptor nucleotide selection. Moreover, a conserved (R/Q)xW motif at the C-terminal loop of CdnEs is proposed as lid residue 3 (lid 3), dictating donor nucleotide specificity to pyrimidine. According to the substrate-bound and intermediate-bound CdnE structures, a synthetic pathway for cyclic di-UMP is deduced. Finally, the rules controlling donor nucleotide specificity of CD-NTases are further demonstrated. This study aids in our understanding of the biochemical mechanisms underlying cyclic di-pyrimidine biosynthesis and provides rules for substrate specificity determination of CD-NTases.

## Results

### *Ef*CdnE synthesizes products cUU and cUC

To understand the molecular basis for synthesizing cyclic di-pyrimidines, the substrate specificity of *Ef*CdnE was first accessed by ITC. The binding of AMPcPP and GMPcPP to *Ef*CdnE was too weak to be detected (Supplementary Fig. 1A, B), indicating that *Ef*CdnE prefers pyrimidine nucleotides over purine nucleotides. The binding profile of UMPnPP to *Ef*CdnE indicates that one *Ef*CdnE protein molecule binds to two UMPnPP molecules, one in the donor pocket and the other in the acceptor pocket, with two different dissociation constants ($K_D$) of $1.47 \times 10^{-8}$ M and $9.8 \times 10^{-7}$ M, respectively (Supplementary Fig. 1C). To investigate whether *Ef*CdnE could distinguish UTP from CTP, titration

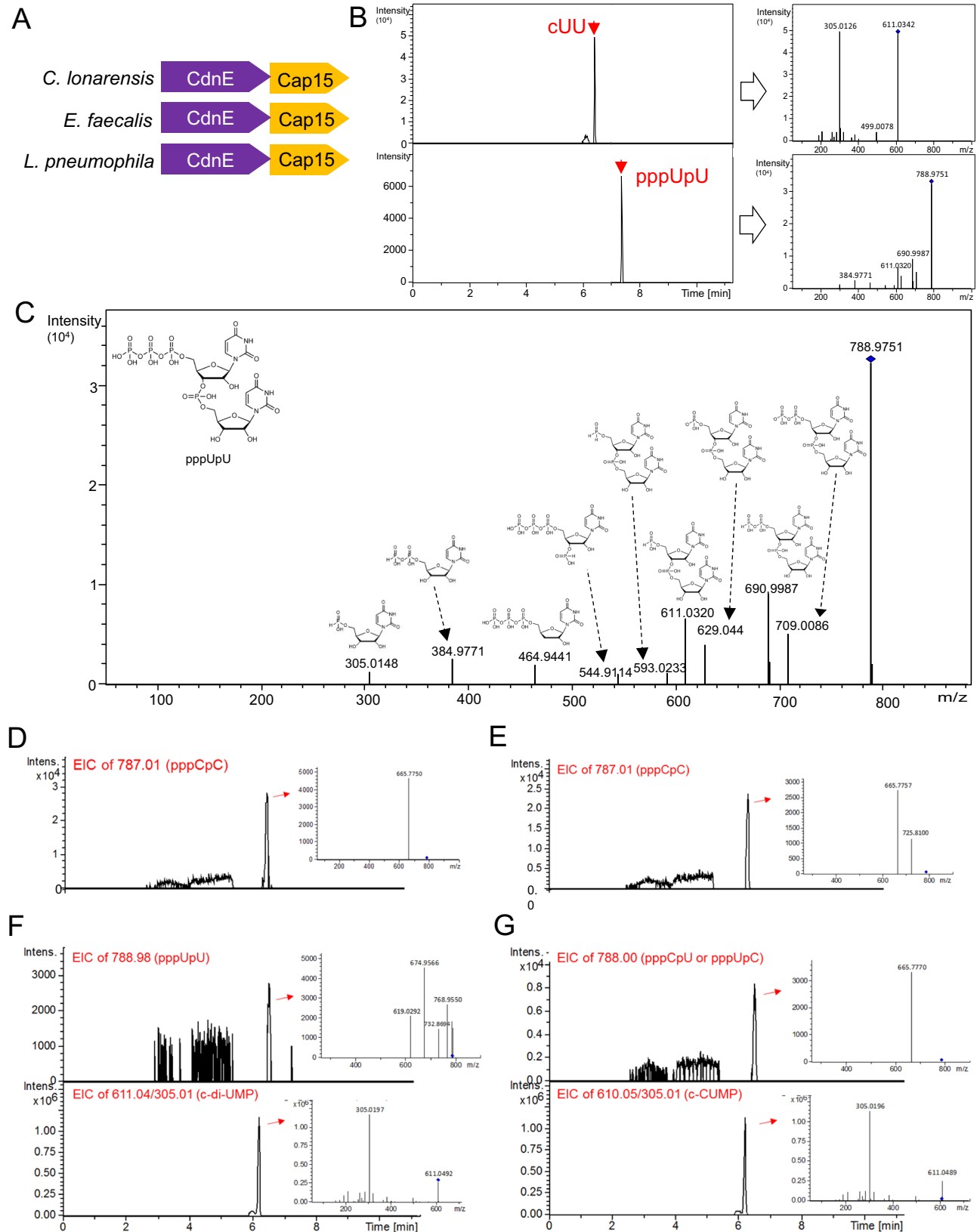

with CMPcPP was conducted. One *Ef*CdnE protein molecule also binds to two CMPcPP molecules, and the $K_D$ values of $1.87 \times 10^{-8}$ M and $1.27 \times 10^{-6}$ M, respectively, are slightly larger than those of UMPnPP (Supplementary Fig. 1D). CMPcPP did not bind to *Ef*CdnE, which was pre-incubated with UMPnPP (Supplementary Fig. 1E), suggesting that the binding of UMPnPP to *Ef*CdnE could exclude the binding of CMPcPP.

To understand the products synthesized by *Ef*CdnE, liquid chromatography with tandem mass spectroscopy (LC-MS/MS) experiments were performed. After overnight incubation of purified *Ef*CdnE with UTP and Mg²⁺, the elution peaks corresponding to product cUU and intermediate pppUpU were detected (LC-MS dataset, Fig. 2B and Supplementary Fig. 2). Different fragments derived from pppUpU,

**Fig. 2 | LC-MS/MS analysis of the reaction intermediate and products synthesized by *Ef*CdnE. A** A schematic of the full CBASS operons for *Cl*CdnE, *Ef*CdnE, and *Lp*CdnE. The effector Cap15 contains two transmembrane segments and a β-barrel and has been shown to cause inner membrane disruption upon ligand binding and oligomerization[20]. **B** LC-MS/MS analysis of the reaction mixture containing purified *Ef*CdnE, UTP and Mg²⁺ ion after overnight incubation. Top panel, a peak of precursor ion (m/z 611.0342) and its fragmented ions (m/z 305.0126 and m/z 499.0078) was detected at 6.4 min, which correspond to cUU according to the matched LC retention time and MS and MS/MS spectra with the chemical standard. Bottom panel, a peak of precursor ion (m/z 788.9751) and its numerous fragmented ions (see below) was detected at 7.4 min, which correspond to pppUpU. **C** The enlarged view of the mass spectrum shown in bottom panel in (**A**). The fragmented ions (m/z 709.0086, m/z 690.9987, m/z 629.044, m/z 611.032, m/z 593.0233, m/z

544.9114, m/z 464.9441, m/z 384.9771, m/z 305.0148) and their corresponding chemical structures derived from pppU[3′–5′]pU were shown. **D** Mass spectrum from a sample of *Ef*CdnE incubated with CTP. Extracted ion chromatograms (EIC) of a fragment ion (m/z 787.01), corresponding to pppCpC. **E–G** Mass spectrum from a sample of *Ef*CdnE incubated with CTP and UTP. **E** EIC of a fragment ion (m/z 787.01), corresponding to pppCpC. **F** Top panel, EIC of a fragment ion (m/z 788.98), corresponding to pppUpU. Bottom panel, EIC of fragment ions of m/z 611.04 and m/z 305.01, corresponding to cUU. **G** Top panel, EIC of a fragment ion (m/z 788.00), corresponding to pppUpC or pppCpU. Bottom panel, EIC of fragment ions of m/z 610.05 and m/z 305.01, corresponding to cUC. Some signals before the actual peaks corresponding to intermediate pppUpU, pppCpC, and pppUpC/pppCpU in (**D–G**) could be attribute to noise and it shows that the abundancy of these species is low.

corresponding to UMP, UDP, UTP, pppUp, pUpU, and ppUpU, were also detected (LC-MS dataset, Fig. 2C). By including CTP in the reaction mixture with *Ef*CdnE after overnight incubation, only the pppCpC reaction intermediate was detected (LC-MS dataset, Fig. 2D). Using CTP and UTP as substrates, *Ef*CdnE generated intermediates with different base compositions, pppCpC, pppUpU, and pppUpC/pppCpU and products cUU and cUC (LC-MS dataset, Fig. 2E–G). Because the molecular masses of pppUpC, pppCpU, and the fragments derived from them are identical (LC-MS dataset, Supplementary Fig. 3), we could not determine which intermediate was generated by *Ef*CdnE. Altogether, the above results indicate that *Ef*CdnE catalyzes the synthesis of cUU products through pppUpU intermediate and cUC products through pppUpC or pppCpU as an intermediate.

### Overall structure of cyclic di-pyrimidine-synthesizing CdnE

The native structure of *Cl*CdnE was refined to 2.2-Å resolution (Supplementary Table 1). Like other CD-NTases, the *Cl*CdnE protein folds into two distinct domains (Fig. 3A). The NTase domain contains a central β-sheet composed of five strands: βA–βE. The longest strand, βE in the middle, extends into a flanking β-sheet with strand βB′ on one side and βF and βG on the other. There are two helices, α2, and α3, in this domain. The helical domain consisted of six major helices, including the first helix, α1. CD-NTases are classified as type-I and type-II according to their requirement for an activation mechanism[16]. In type-I enzymes such as cGAS, the NTase and helical domains are connected by the "spine" helix, which is broken into two upon activation[4]. The separation of helices α1 and α2, which correspond to the "spine" in cGAS by a short loop in *Cl*CdnE, may account for its constitutive NTase activity as a type II enzyme.

The native structure of *Ef*CdnE was determined to be a 1.75-Å resolution (Supplementary Table 2). When compared with *Cl*CdnE, helix α2 of *Ef*CdnE was shorter by one turn at the N-terminus (Supplementary Fig. 4). The overall root-mean-square deviation (RMSD) is 1.38 Å between 280 matched pairs of Cα atoms. Other significant deviations were observed in the βB′-α3 loop, βD-βE loop, βE-βF loop, and α7-α8 loop. The native structure of *Lp*CdnE was refined to 2.46-Å resolution (Supplementary Table 3). It differs from *Cl*CdnE in similar regions, with 1.32 Å RMSD for 276 Cα pairs, but the α1-α2 loop is missing (Supplementary Fig. 4). Generally, regions with larger deviations were correlated with higher temperature factors. Despite the differences in local regions, the dispositions of the NTase and helical domains were similar in the native structures of *Cl*CdnE, *Ef*CdnE, and *Lp*CdnE, respectively.

Despite larger than 25% sequence identity (Fig. 3B, C), the structures of *Em*CdnE and *Rm*CdnE showed remarkable differences compared with *Cl*CdnE (Supplementary Fig. 5). Strand βB′ and the entire connection to helix α3 are missing in *Em*CdnE, whereas the βB′-α3 loop in *Rm*CdnE includes a short two-turn helix. The βE–βF loop conformations also varied greatly (Supplementary Fig. 5). The RMSD of the *Cl*CdnE, *Ef*CdnE, and *Lp*CdnE structures from *Em*CdnE and *Rm*CdnE range between 1.50 Å and 1.83 Å for 244 to 271 Cα pairs (Supplementary Table 4).

### UTP substrate binding modes of CdnE

By including substrate UTP or UMPnPP and cofactor MgCl₂ in the crystallization solutions, we obtained the complex crystals of *Cl*CdnE-UTP, *Lp*CdnE-UTP, and *Lp*CdnE-UMPnPP and refined them to 2.6-Å, 1.95-Å and 2.2-Å resolution, respectively (Supplementary Fig. 6 and Supplementary Tables 1 and 3). The RMSD between the 297 pairs of protein Cα atoms in the native and complex structures of *Cl*CdnE was 0.325 Å, and that between the 306 Cα pairs in the native and complex *Ef*CdnE crystals was 0.253 Å (Supplementary Figure 7). The four protein models, two from *Lp*CdnE-UTP and two from *Lp*CdnE-UMPnPP complex show RMSD of 0.56 Å to 0.82 Å from the native *Lp*CdnE for 281 to 286 matched Cα pairs (Supplementary Table 5). The largest deviation is seen in the βB′-α3 loop, probably because of its flexible nature and outward location (Supplementary Fig. 7). The little structural differences from native proteins suggest that substrate binding does not induce large domain movement in all three CD-NTases studied here.

In the *Cl*CdnE-UTP complex structure, the triphosphate part of the donor UTP is bound to the enzyme via a Mg ion, termed Mg-B, as observed for ATP in other CD-NTases[13]. The Mg-B ion, with an octahedral configuration, was coordinated to all three phosphate groups of the UTP, two water molecules, and the side-chain of Asp67 (Fig. 4A). The β-phosphate group is hydrogen-bonded to the backbone of Ser53 and the side-chain of Arg197. The side-chain of Arg61 lies adjacent to γ-phosphate, but no direct bonding is observed. The ribose moiety of the donor UTP is not involved in direct bonding to the protein. However, its pyrimidine base forms stacking interactions with the side-chain of Phe217 on one side and the base of the acceptor UTP on the other. A hydrogen bond between the α-phosphate of the donor UTP and ribose O3′ of the acceptor was also present. The triphosphate moiety of the acceptor UTP forms two salt bridges on the side-chains of Lys123 and Lys276, respectively (Fig. 4B). The weaker interactions may correlate with its higher flexibility, as implied by the presence of electron density for the alternative disposition of the γ-phosphate group. In contrast, the base of the acceptor UTP is engaged in three well-defined hydrogen bonds with the protein side chains: one with Gln51 and two with Asn169. The proximity of the Phe159 side-chain suggested a T-stacking interaction with the acceptor base. In addition to the absence of an acceptor-bound Mg ion, the arrangement of UTP molecules is consistent with the canonical binding mode for the NTase superfamily[17].

In the *Lp*CdnE-UTP complex crystal, similar binding modes of the donor and acceptor nucleotides to the active site were observed for both protein molecules. In addition to the Mg-B ion, the α-phosphate of the donor UTP also forms a coordinate bond with a Mg ion, Mg-A, which further shares the Asp67 side-chain and a water molecule with Mg-B in octahedral-complex formation. The other ligands are the Asp139 side-chain, the ribose O3′ of the acceptor nucleotide, and a water molecule. The γ-phosphate group of the donor formed four hydrogen bonds with Ser53, Asn59, Lys201, and Ser220 of the protein in *Lp*CdnE (Fig. 4C, E), showing remarkably stronger interactions than those in *Cl*CdnE and *Ef*CdnE. However, the stacking between the Tyr221 side-chain and the donor base may be weaker. The interaction

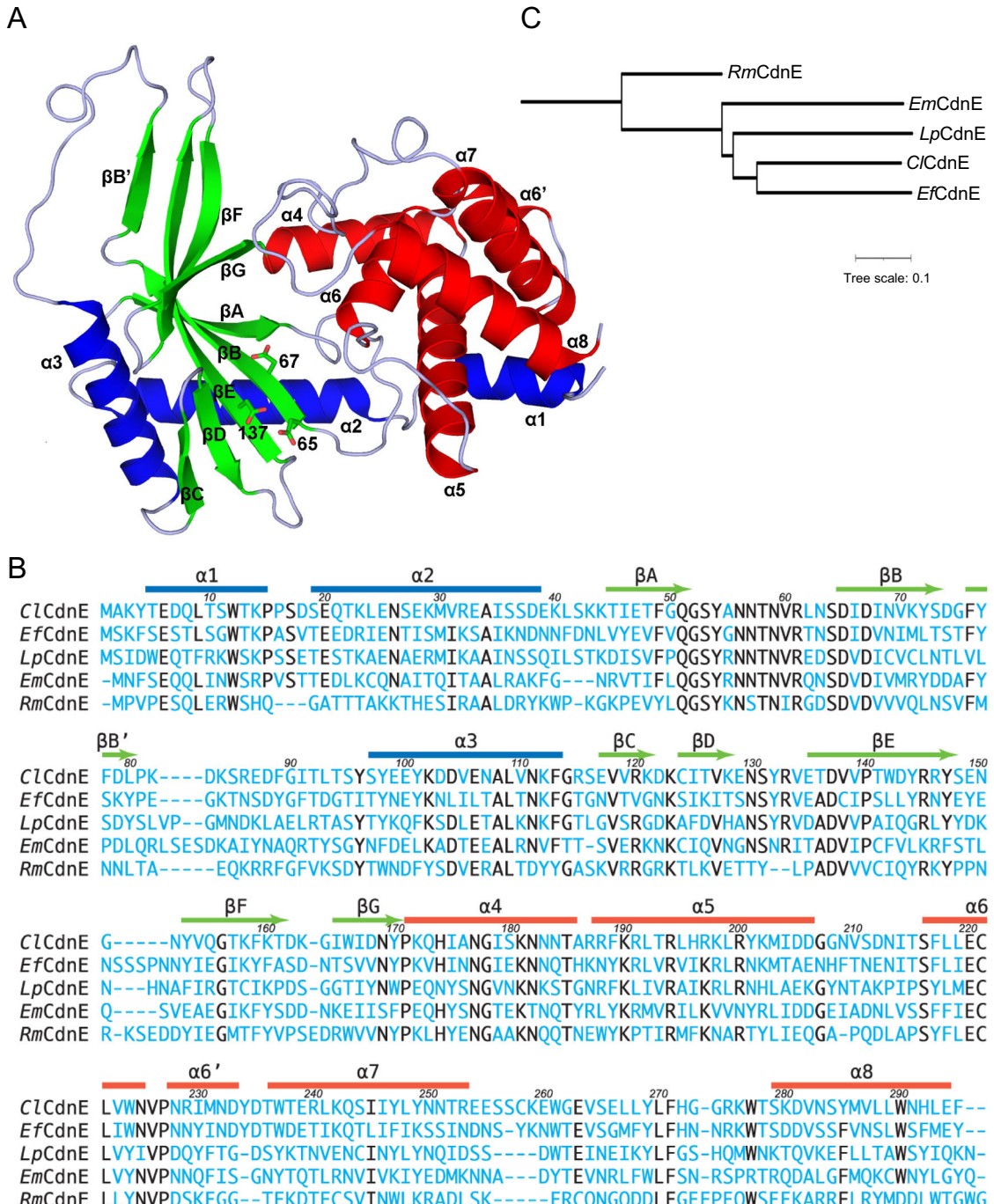

**Fig. 3 | Overall structure of cyclic di-pyrimidine-synthesizing CdnE. A** In this Ribbons diagram of *Cl*CdnE, the N- and C-terminal α-helices are colored blue and red, and the β-strands are green. The N-terminal NTase domain on the left side contains the three conserved catalytically important aspartate residues, which are shown as sticks. The first helix α1, disconnected from helix α2 by a short loop, associates with the C-terminal helical domain on the right. **B** The amino-acid sequences of *Cl*CdnE, *Ef*CdnE and *Lp*CdnE in this study are aligned with those of *Em*CdnE (PDB 6E0M) and *Rm*CdnE (PDB 6E0K) based on their tertiary structures. The corresponding secondary structure elements are represented as bars for α-helices and arrows for β-strands on the top, which are colored similarly as in (**A**).

Conserved amino acids are emphasized by using black letters. The residue numbers above the sequences are from *Cl*CdnE. **C** Phylogenetic tree of *Cl*CdnE, *Ef*CdnE, *Lp*CdnE, *Em*CdnE and *Rm*CdnE. The scale bar corresponds to a phylogenetic distance of 0.1 nucleotide substitution per site. *Cl*CdnE was most similar to *Ef*CdnE, with 44% identity. Both *Cl*CdnE and *Ef*CdnE are more similar to *Em*CdnE, which turns out di-purine products cGA, cAA, and cGG, with 33% sequence identity to the cUA-producing *Rm*CdnE with 25% and 27% identity, respectively. *Lp*CdnE, which makes cUU, shows 36% and 34% sequence identity to *Cl*CdnE and *Ef*CdnE, respectively, which is slightly higher than the 32% and 31% identity to *Em*CdnE and *Rm*CdnE, respectively.

was also weaker between the non-aromatic Ile163 side-chain and the acceptor base, which forms similar hydrogen bonds to the side-chains of Gln51 and Asn173, as seen in *Cl*CdnE and *Ef*CdnE (Fig. 4D, F). Besides the bond of O3′ to Mg-A, the acceptor ribose has O2′ hydrogen bonded to the Asp67 side-chain. *Lp*CdnE binds more strongly to the acceptor triphosphate than *Cl*CdnE. Although there is some disorder in the β- and γ-phosphate groups, the triphosphate is with hydrogen bond to the backbone N of Asp124, Lys125, and Ala126 in the βC–βD loop, and salt bridged to the Lys125 side-chain. In one of the conformations, β-phosphate forms a hydrogen bond with ribose O3′ (Fig. 4F).

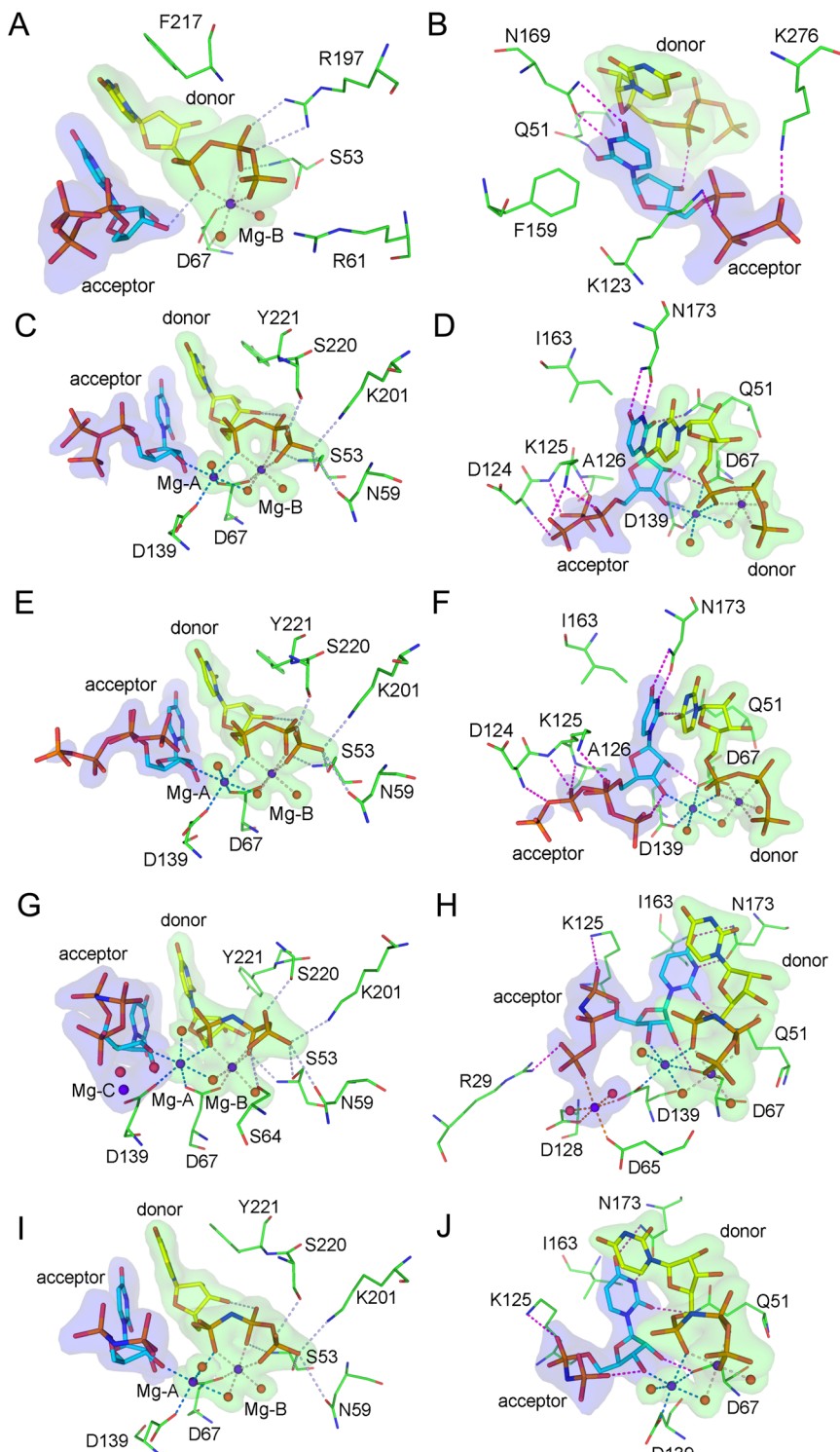

**Fig. 4 | Substrate binding modes in *Cl*CdnE and *Lp*CdnE.** The bound nucleotides are shown as thick stick models and the amino acid residues as thin sticks. The magnesium ion and the associated water molecules are shown as spheres. Hydrogen bonds and coordinate bonds are indicated by using dashed lines. The models are superimposed on the *Fo-Fc* maps each calculated by omitting the atoms in question and contoured at 4-σ level. In (**A**), the view is centered on the donor nucleotide in the *Cl*CdnE-UTP complex; in (**B**) it is centered on the acceptor nucleotide. **C–F** Substrate binding modes in *Lp*CdnE. The monoclinic crystal contains two protein molecules in an asymmetric unit. **C, D** One of them centered on the donor and acceptor nucleotides, respectively. **E, F** show another. **G–J** Binding modes of the substrate analog in *Lp*CdnE. The orthorhombic crystal contains two protein molecules in an asymmetric unit. (**G**) and (**H**) show one of them centered on the donor and acceptor binding sites, respectively. **I, J** show another.

In the *Lp*CdnE-UMPnPP complex crystal, the donor nucleotide makes nearly identical interactions with the enzyme as in the UTP complex (Fig. 4G, I), except for an additional bond between a β-phosphate and the Ser64 side-chain (Fig. 4G). The base and ribose parts of the acceptor UMPnPP were also engaged in very similar interactions with the protein as those of the UTP complex (Fig. 4H, J). However, its triphosphate-analog part does not interact extensively with the backbone atoms in the βC-βD loop. Only a salt bridge was

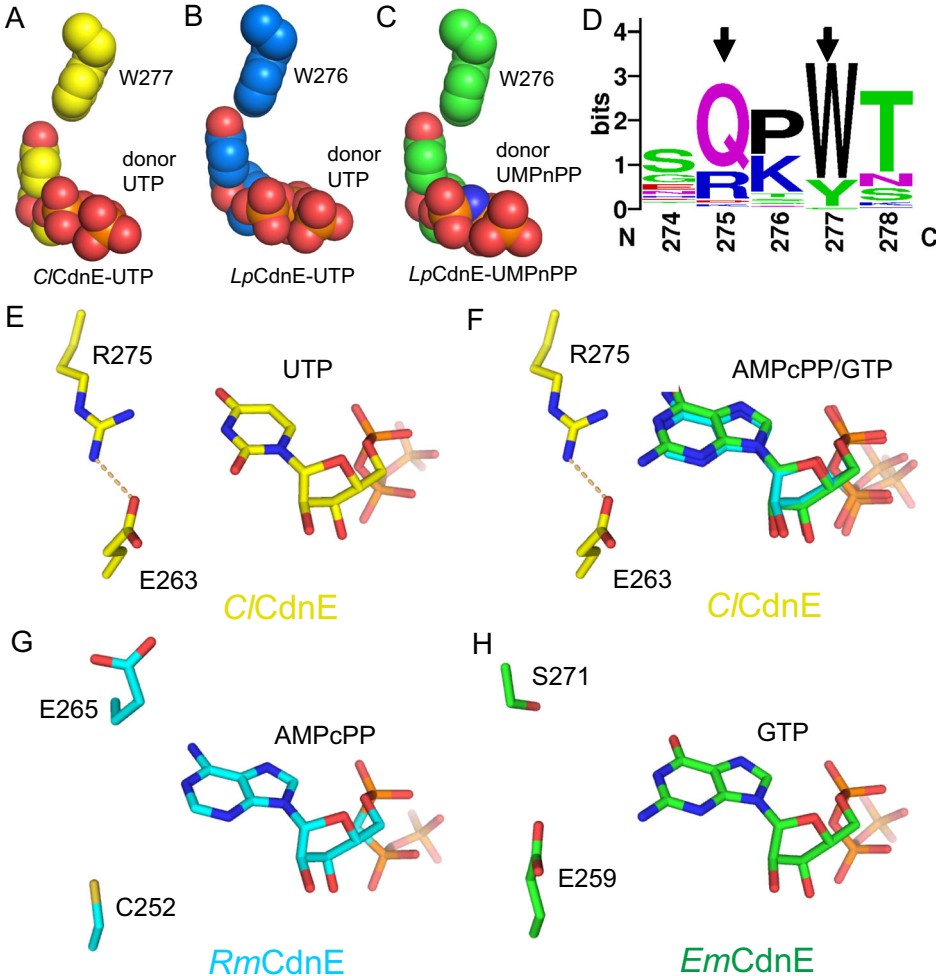

**Fig. 5 | The structural determinants for donor nucleotide specificity of cyclic di-pyrimidine-synthesizing CdnE. A–C** The conserved Trp residue in (**A**) *Cl*CdnE-UTP, (**B**) *Lp*CdnE-UTP, (**C**) *Lp*CdnE-UMPnPP that makes direct contact with the uracil base of donor nucleotide. The ligand and sidechain of Trp are shown in spheres. **D** The sequence logo of the identified lid 3 of cyclic di-pyrimidine-synthesizing CdnE. The residue number is based on *Cl*CdnE. The determinants for pyrimidine specificity are indicated by black arrows. **E–H** The complex structures of *Cl*CdnE, *Rm*CdnE (PDB 6EOL) and *Em*CdnE (PDB 6EON) are superimposed while focusing on the donor nucleotides. In (**E**) and (**F**) the side-chains of Glu263 and Arg275 of *Cl*CdnE form a salt bridge (yellow dashed line) that restrict the space for nucleobase. Uracil (and probably cytosine) fits snugly (**E**) but adenine and guanine can cause significant clashes (**F**). In *Rm*CdnE the corresponding residues of Cys252 and Glu265 are well separated, making a more open space for adenine base (**G**). Similarly, the side-chains of Glu259 and Ser271 in *Em*CdnE are not associated, allowing the binding of guanine base (**H**).

observed between α-phosphate and Lys125 side-chains. One of the *Lp*CdnE-UMPnPP complexes contained a third Mg ion, Mg-C (Fig. 4G). It is coordinated by the side chains of Asp65, Asp128, and Asp139, two water molecules, and the γ-phosphate of the acceptor UMPnPP, which forms a salt bridge with the Arg29 side-chain (Fig. 4H). Although Asp65 and Asp139 are conserved, Arg29 and Asp128 are not. Mg-C appears accidental but may also be related to the catalytic pathway. After the first step of the reaction, the resulting pyrophosphate dissociates from the donor nucleotide as a Mg-complex, and the acceptor triphosphate moiety, now part of the intermediate, may occupy the donor-binding site more readily along with the bound Mg-C than without a pre-bound Mg ion. In DNA/RNA polymerases, the 3' end of the primer substrate is tightly bound to the associated metal A, and the metal B containing the donor nucleotide shows weaker binding[21]. In contrast, CD-NTases appear to bind more strongly to metal B. In the complex structures of *Cl*CdnE and *Ef*CdnE described above, as well as in the three previously solved complex structures of *Em*CdnE (PDB 6EON and 6EOO, respectively)[12] and *Rm*CdnE (PDB 6EOL)[12], only Mg-B, but not Mg-A, was observed. However, both Mg-A and Mg-B were observed in the complex structure of *Lp*CdnE.

Analyses of the Mg-binding modes show that the distances from the respective ligand atoms are 2.2-2.5 Å for Mg-A, 2.0-2.2 Å for Mg-B, and 2.1-2.3 Å for Mg-C (Supplementary Table 6). The longer distances to Mg-A are consistent with its weaker binding, but this might indicate the use of other metal ions, such as manganese.

**Conserved motif dictates donor pyrimidine selection of CdnE**
The substrate-bound *Cl*CdnE and *Lp*CdnE presented here showed that recognition of acceptor UTP or UMPnPP is dictated by conserved Asn169 and Asn173 (lid 1), respectively, consistent with a previous report[12]. In contrast, the side-chain of Asn178 (lid 2) of *Lp*CdnE forms a hydrogen bond only with the ribose 2'-OH of the donor UTP/UMPnPP, and His174 (lid 2) of *Cl*CdnE makes no contact with donor UTP (Supplementary Figure 8), suggesting that lid 2 may plays a role in nucleotide specificity during catalysis or through some other mechanism. Instead, Trp277 and Trp276 form extensive hydrophobic contacts with the base moiety of donor UTP/UMPnPP in the substrate-bound *Cl*CdnE and *Lp*CdnE, respectively (Fig. 5A–C). Moreover, superimposition of the *Cl*CdnE complex structure with those of *Rm*CdnE and *Em*CdnE revealed that the bound AMPcPP and

GTP could not be well accommodated in the donor site of *Cl*CdnE, but would cause significant clashes with the side-chains of Glu263 and Arg275, which come together to form a salt bridge (Fig. 5E, F). Similarly, the side chains of Glu267 and Arg279 in *Ef*CdnE are engaged in salt-bridging interactions and would have substantial clashes with the donor base if it were a purine nucleotide. This observation explains why we could not detect the binding of ATP and GTP analogs to *Ef*CdnE in the ITC experiments. In *Lp*CdnE, the Glu262 and Gln274 side chains also come together to form a hydrogen bond and restrict the donor-binding site for pyrimidine nucleotides. The corresponding Cys252 and Glu265 in *Rm*CdnE or Glu259 and Ser271 in *Em*CdnE were well separated (Fig. 5G, H), and thus a purine base can be accommodated. Sequence analysis of cyclic di-pyrimidine-synthesizing CdnE proteins further revealed the conservation of Arg/Gln and Trp residues, together named the (R/Q)xW motif (Fig. 5D), which restricts the donor pocket and forms direct hydrophobic contacts with the uracil base of donor nucleotides, as mentioned above.

To validate the importance of this motif, we changed Arg279 and Trp281 of *Ef*CdnE to a smaller alanine residue and assessed the substrate specificity and product synthesis of both wild-type and mutant *Ef*CdnE by ITC and LC-MS/MS. The Arg-to-Ala reprogramming of *Ef*CdnE restored its binding to both ATP and GTP with affinities near micromolar dissociation constants (Fig. 6A, B and Supplementary Figure 9A), in contrast to the lack of binding of purine nucleotides by wild-type *Ef*CdnE (Supplementary Fig. 1A, B). Similarly, *Ef*CdnE[W281A] binds AMPcPP with a $K_D$ value of $3.5 \times 10^{-5}$ M and GMPcPP with a $K_D$ value of $6.5 \times 10^{-5}$ M (Fig. 6C and Supplementary Fig. 9B), indicating that mutating either Arg or Trp of the (R/Q)xW motif is sufficient to rewire the specificity of the donor pocket of *Ef*CdnE from pyrimidine to purine. LC-MS/MS analysis further demonstrated that incubation of *Ef*CdnE[R279A] or *Ef*CdnE[W281A] with UTP/ATP and UTP/GTP mixtures yielded products corresponding to cUA and cUG, respectively (LC-MS dataset, Fig. 6D, E and Supplementary Fig. 9C–F). In conclusion, changing one of the two residues in the (R/Q)xW motif switches the product synthesis of *Ef*CdnE from cyclic di-pyrimidine to purine-pyrimidine mixed species of CDNs, validating the critical role of the (R/Q)xW motif in pyrimidine selection. Based on sequence conservation and structural similarity, it is proposed that the cyclic di-pyrimidine-synthesizing CdnE proteins, including *Cl*CdnE, *Ef*CdnE, and *Lp*CdnE, form smaller donor pockets that act as molecular gauges to recognize and retain the binding of pyrimidine nucleotides but weaken or exclude the binding to purine nucleotides (Fig. 6F). In combination with the conserved Asn residue in the acceptor pocket, the substrate specificity for double pyrimidine was achieved (Fig. 6F).

### *Ef*CdnE synthesizes 2′3′-cUU as major product

We showed that *Ef*CdnE could use UTP as a substrate to produce cUU, using pppUpU as an intermediate (Fig. 2A). The *Ef*CdnE-pppU[3′–5′]pU complex structure presented here shows that the triphosphate moiety of the reaction intermediate is coordinated to the Mg-B ion like that of the donor nucleotide in *Cl*CdnE. However, there was only one hydrogen bond between the β-phosphate and the Ser53 side-chain (Fig. 7A). The side-chain of Lys202 is more than 4.5 Å from the triphosphate, while Arg61 is farther away. The uracil base in the donor binding site was flipped over, and the ribose was turned around compared with the donor structure in the *Cl*CdnE-UTP complex (Fig. 7B). Neither part directly interacts with the protein, except for stacking the base with the Phe222 side-chain. The acceptor-base forms hydrogen bonds with the Gln51 and Asn174 side-chains in a similar manner as their equivalents in *Cl*CdnE and is packed against the aromatic side-chain of Tyr164 (Fig. 7C). The acceptor ribose forms a hydrogen bond with the Asp67 side-chain via its 2′-OH group and an O3′ hydrogen bond to the donor α-phosphate. The phosphate group linking the donor and acceptor did

not interact directly with the protein, despite the presence of a nearby Lys123 side-chain.

Including UTP and MgCl$_2$ yields another complex crystal of *Ef*CdnE, in which an intermediate pppU[2′–5′]p binds to the ligand-binding pocket. The complex structure was refined to a 2.0-Å resolution (Supplementary Table 2). The overall structure of *Ef*CdnE-pppU[2′–5′]p is nearly identical to *Ef*CdnE-pppU[3′–5′]pU, with an RMSD of 0.159 between 280 matched Cα pairs. The intermediate pppU[2′–5′]p, devoid of the acceptor base, binds to the donor pocket with an additional phosphate linked to the ribose 2′-O. The triphosphate part of the intermediate pppU[2′–5′]p is similarly coordinated to the Mg-B ion as that of pppU[3′–5′]pU (Fig. 7A, D). The β-phosphate forms two hydrogen bonds with the main- chain and side-chain of Ser53 (Fig. 7D). The side-chain of Ser221 is near γ-phosphate, forming a weak hydrogen bond (3.8 Å) (Fig. 7D). The side-chain of Phe222 forms a stacking interaction with the donor uracil base, similar to *Cl*CdnE and *Lp*CdnE (Figs. 4 and 7D). The ribose 2′-O-linked phosphate forms two hydrogen bonds with the side-chain of Lys123 (Fig. 7D). The side-chain of conserved R279 directly hydrogen bonded to the 2′-O of the uracil base, supporting its importance in pyrimidine selection in the donor pocket (Fig. 7D). LC-MS/MS detected the formation of pppUpU as an intermediate by *Ef*CdnE upon UTP was used as a substrate (Fig. 2A) but could not distinguish the formation of 2′–5′ or 3′–5′ phosphodiester linkages. Here, we identified two kinds of intermediates, pppU[3′–5′] pU and pppU[2′–5′]p, in the crystal structures of *Ef*CdnE by incubating the proteins with UTP and MgCl$_2$, suggesting that the first step of the nucleotidyl-transfer reaction proceeds by attacking α-phosphate by either 2′-OH or 3′-OH of the acceptor UTP. However, superimposition of *Ef*CdnE-pppU[3′–5′]pU with *Ef*CdnE-pppU[2′–5′]p demonstrated that the terminal uracil base of pppU[2′–5′]pU could not fit into the acceptor pocket. It was probably too flexible to be modeled in the structure, suggesting that the re-orientation and re-binding of pppU[2′–5′]pU into the active site is difficult and energetically unfavored (Fig. 7E). Furthermore, our determined substrate-bound *Cl*CdnE and *Lp*CdnE showed that the 3′-OH is much closer to the α-phosphate than the 2′-OH of the acceptor UTP/UMPnPP (Supplementary Fig. 10A–C), suggesting the formation of pppU[3′–5′]pU as an intermediate. Moreover, in *Ef*CdnE-pppU[3′–5′]pU complex structure, the 2′-OH seems to be positioned ready for attacking (3.5 Å) instead of 3′-OH (5.1 Å, Supplementary Figure 10D), which will lead to the production of 2′3′-cUU. Based on the above structural observations, we proposed a 2′3′-cUU/3′3′-cUU synthetic pathway involving *Ef*CdnE (Fig. 7F). The first nucleotidyl-transfer reaction is catalyzed by the attack of the α-phosphate of the donor nucleotide by the acceptor ribose 3′-OH (Fig. 7F). Our determined *Ef*CdnE complex with two different intermediates supports the synthesis of pppU[3′–5′]pU rather than pppU[2′–5′]pU; the latter could not fit well into the active site of *Ef*CdnE for the subsequent reaction. Cyclization of the intermediate seems to favor the formation of a 2′–5′ linkage over the 3′–5′ linkage, judging by the *Ef*CdnE-pppU[3′–5′]pU complex structure. Therefore, 2′3′-cUU instead of 3′3′-cUU is assumed to be the major product synthesized by *Ef*CdnE (Fig. 7F).

## Discussion

Altogether, the structural and functional studies of *Ef*CdnE presented here prove that it is an authentic cyclic di-pyrimidine-synthesizing CD-NTase capable of simultaneously binding to two molecules of pyrimidine nucleoside triphosphate and catalyzing their cyclization to produce cyclic di-pyrimidine nucleotides. Based on its extensive structural similarity, *Cl*CdnE probably functions similarly. Previous studies have demonstrated that the conserved asparagine residue (e.g., Asn166 in *Rm*CdnE) in the active-site lid is important for uracil base recognition of the acceptor nucleotide-binding pocket of CD-NTases[12,19]. Here, the structures of three cyclic di-pyrimidine-synthesizing CD-NTases, *Cl*CdnE, *Ef*CdnE, and *Lp*CdnE, confirmed

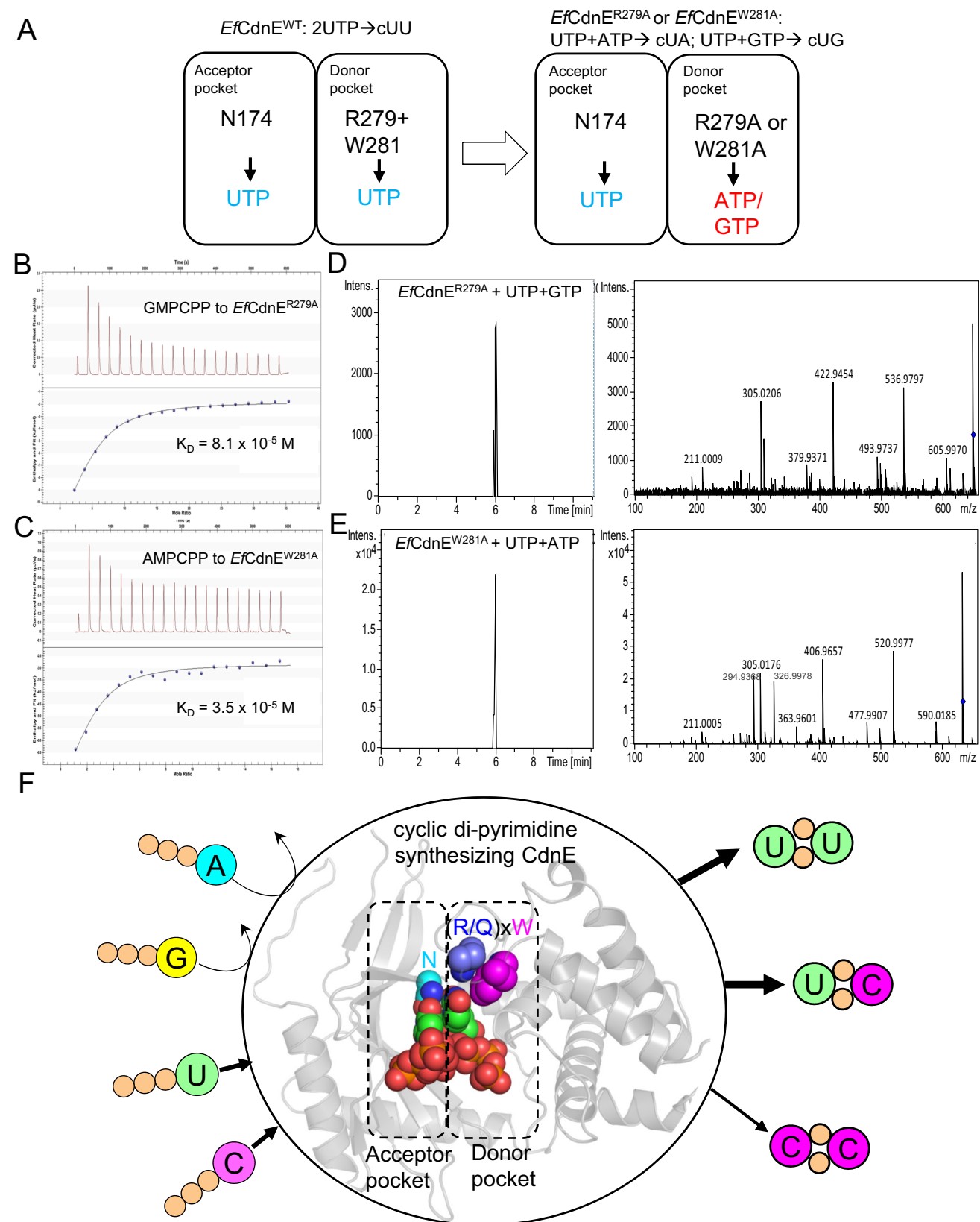

this key finding. In structural terms, by flipping the side-chain amide group of this conserved asparagine residue, a cytosine base can also fit into the pocket with proper hydrogen bond formation (Supplementary Figure 11), explaining why both UTP and CTP can bind to these enzymes.

The similar binding affinity of UMPnPP and CMPcPP to *Ef*CdnE suggests that cyclic di-pyrimidine- synthesizing CdnE distinguish the smaller six-membered pyrimidine bases from the larger nine-membered purine bases instead of recognizing the subtle structural differences in the nucleobases. If any of the uracil bases in the

**Fig. 6 | Functional validation of the critical role of (R/Q)xW motif of cyclic di-pyrimidine-synthesizing CdnE. A** Schematic representation of the mutational strategy for re-wiring the substrate and product specificity. ITC analysis of the interaction between (**B**) non-hydrolysable GMPcPP and *Ef*CdnE^R279A and (**C**) non-hydrolysable AMPcPP and *Ef*CdnE^W281A. The determined dissociation constants (K_D, M) were indicated. The experiments were repeated at least twice. LC-MS/MS analysis of the products synthesized by incubating (**D**) *Ef*CdnE^R279A with 1:1 UTP/GTP mixture and (**E**) *Ef*CdnE^W281A with 1:1 UTP/ATP mixture. In (**D**), a peak of m/z 649.01

and its fragmented ions was found to be eluted at 6.1 min having the same elution time of cyclic UMP-GMP. In (**E**), a peak of m/z 633.03 and its fragmented ions was found to be eluted at 6.1 min having the same elution time of cyclic UMP-AMP.
**F** The cartoon model (based on *Cl*CdnE-UTP complex) showing the substrate and product preference of cyclic di-pyrimidine-synthesizing CdnE. The conserved (R/Q) xW motif in donor pocket and conserved asparagine residue (N) plus two donor UTP and acceptor UTP were shown in spheres. The substrates can be UTP or CTP. The products were identified to be cUU, cUC and cCC.

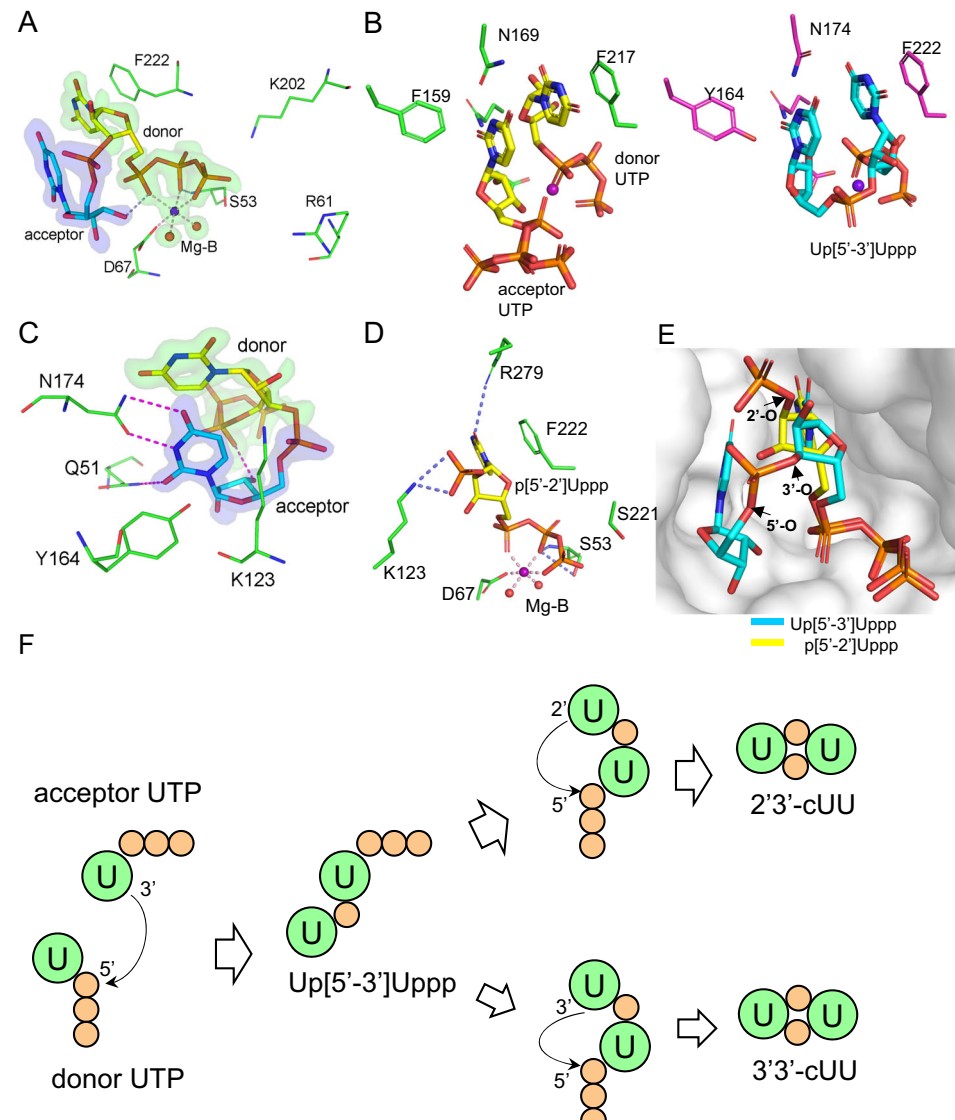

**Fig. 7 | *Ef*CdnE produce 2'3'-cUU as major product through formation of intermediate pppU[3′−5′]pU. A, C** The *Ef*CdnE-pppU[3′−5′]pU complex model, centered on the donor and acceptor parts, respectively. **B** Structural comparison of *Cl*CdnE-UTP (left) and *Ef*CdnE-pppU[3′−5′]pU (right). The donor and acceptor UTP from the *Cl*CdnE-UTP complex have their carbon atoms colored yellow; pppU[3′−5′] pU from the *Ef*CdnE-pppU[3′−5′]pU complex have cyan carbons. The protein side-chains are colored in green (*Cl*CdnE) and magenta (*Ef*CdnE), respectively. **D** The

*Ef*CdnE-pppU[2'-5']p complex model, centered on the donor part.
**E** Superimposition of *Ef*CdnE-pppU[3′−5′]pU and *Ef*CdnE-pppU[2′−5′]p complex models. pppU[3′−5′]pU and pppU[2′−5′]p have cyan and yellow carbons, respectively. The ligand-binding pocket is shown in surface presentation. **F** The proposed synthetic pathway for 2'3'-cUU by *Ef*CdnE through the formation of intermediate pppU[3′−5′]pU.

complex crystal structures of *Cl*CdnE, *Ef*CdnE, and *Lp*CdnE is replaced by cytosine, the base can still fit well in the active site. With a flipping of the latter amide group, the side-chains of Gln51 and Asn174 in *Ef*CdnE can form proper hydrogen bonds with the O2, N3, and N4 atoms of the cytosine base, and so can the equivalents in *Cl*CdnE and *Lp*CdnE. Although both acceptor and donor sites can

bind to CTP, the preferred substrate is UTP. The cyclization reaction of CTP could not be completed, turning out only the intermediate pppCpC but no cCC product (Fig. 2C). The reason might lie in the binding mode of the donor moiety upon pppCpC rearrangement in the active site. To assume the pppU[3′−5′]pU-like binding mode, the donor base should be flipped over, and the ribose rotated to the

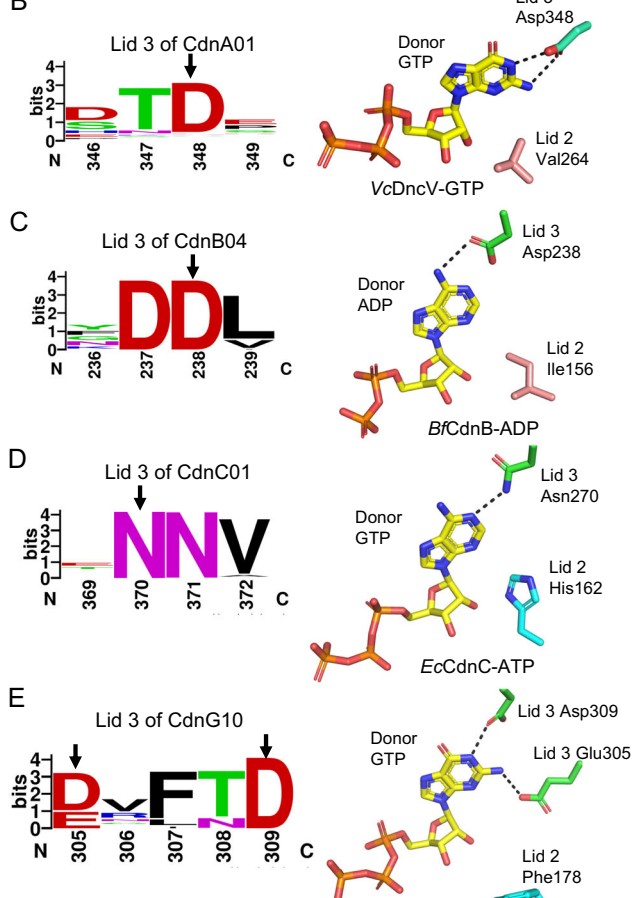

| CD-NTase | Clade | Lid 2 | Lid 3 | Donor substrate |
|---|---|---|---|---|
| *Vc*DncV | A01 | V | D | GTP |
| *Bf*CdnB | B04 | I | D | GTP |
| *Ec*CdnC | C01 | H | N | ATP |
| *Lp*CdnE | E02 | N | (R/Q)xW | UTP |
| *Cl*CdnE | E02 | H | (R/Q)xW | UTP |
| *Ef*CdnE | E02 | H | (R/Q)xW | UTP |
| *Bd*CdnG | G10 | F | DxxxD | GTP |

**Fig. 8 | "Lid 3" dictates the donor nucleotide specificity of CD-NTases. (A)** Summary of the specificity-determining residues for donor nucleotides of CD-NTases in different clades. **(B–E)** Left, the sequence logo of lid 3 of **(B)** CdnA01, **(C)** CdnB04, **(D)** CdnC01 and **(E)** CdnG10. The residues forming direct interaction with donor nucleotide are indicated by arrows. Right, the enlarged view of the donor pocket of the representative of **(B)** CdnA01 (*Vc*DncV, PDB 4XJ3), **(C)** CdnB04 (*Bf*CdnB, PDB 7LJO), **(D)** CdnC01 (*Ec*CdnC, PDB 6P80), **(E)** CdnG10 (*Bd*CdnG, PDB 7LJN). The bound donor nucleotide, lid 2 and 3 are shown in sticks.

AMP produced by CD-NTase in clade D from *Acinetobacter baumannii*, which activates Cap4 effectors for double-stranded DNA degradation, resulting in cell deaths of infected bacteria[22]. The unique 3'2' or 2'3'-phosphodiester linkage of bacterial second messenger may provide additional protection against phosphodiesterase degradation from invading phages[15]. In this study, we provide structural insights into the mechanism underlying synthesizing 2'3'-cUU by *Ef*CdnE, which could be beneficial to the design and engineering of cyclic di-pyrimidine-synthesizing enzymes. However, whether specific isomer of cUU activates the downstream Cap15 effector remains unclear and need to be characterize in the future study.

Inspired by the key finding that the conserved residues at the C-terminal loop above the donor pocket, named lid 3, dictate pyrimidine nucleotide selection, we identified conserved lid 3 at the equivalent position of CD-NTases in clades A, B, C, E, and G, which directly controls donor nucleotide specificity (Fig. 8A). *Vc*DncV belongs to CD-NTase in clade A01 (CdnA01), which produces 3'3'-cGA as the major product and 3'3'-cGG and 3'3'-cAA as the minor products[23]. Based on current available structures, lid 2 of VcDncV (Val264) was found to make van der Waals interaction with the donor nucleotide but does not form base-specific interactions (Fig. 8B). Instead, Asp348 of *Vc*DncV forms bidentate hydrogen bonds with the guanine base (Fig. 8B), which is the major determinant for the high-affinity binding of the GTP substrate[18]. Sequence analysis showed that lid 3 (Asp348) of *Vc*DncV was highly conserved in CdnA01 (Fig. 8B). Similarly, *Bf*CdnB from *Bacteroides fragilis* also contains a highly conserved Asp238 as lid 3 of CD-NTase in clade B04 (CdnB04), which prefers GTP over ATP and synthesizes mainly 3'3'-cGA, similar to *Vc*DncV (Fig. 8C). Furthermore, two highly conserved Asn residues among CD-NTases in clade C01 (CdnC01) were identified as lid 3 (Fig. 8D). One of the conserved Asn residues (Asn270) forms a hydrogen bond with the N1 atom of the adenine base of donor ATP in *E. coli* CdnC (Fig. 8D). Finally, the conserved (D/E)xxxD motif of CD-NTases in clade G10 (CdnG10) was identified as lid 3 (Fig. 8E). The structure of *Bd*CdnG complexed with GTP (PDB 7LJN) demonstrated that the former Glu residue (Glu305) hydrogen bonded to the 2-NH₂ group of the guanine base of donor GTP, while the latter Asp residue coordinated with the guanine N1 atom of donor GTP (Fig. 8E). In summary, the identified conserved lid 3 directly interacts with the donor nucleotide and controls its specificity, unlike the previously proposed lid 2. Our results could help facilitate future studies on CD-NTases and are described below. First, the Asp residue prefers the binding of guanine bases owing to the formation of bidentate hydrogen bonds rather than one hydrogen bond with adenine bases. Second, the conserved Asn residue does not distinguish ATP from GTP; however, the cellular concentration of ATP is much higher than that of GTP, leading to the selective advantage of ATP by Asn residues. Third, combining two adjacent Asp/Glu residues enhances GTP binding.

Little is known about pyrimidine-containing signaling molecules' synthesis and recognition mechanisms in biological systems. For example, cyclic UMP (cUMP) and cyclic CMP (cCMP) are implicated in the regulation of apoptosis, necrosis, immune modulation, and embryonic development in mammalian cells[24] and the induction of the reactive oxygen species (ROS) response in plants[25]. However, the exact synthases and receptors of cUMP and cCMP in eukaryotes remain elusive. cUMP and cCMP are found to mediate the anti-phage defense function in bacteria through Pycsar (pyrimidine cyclase system for antiphage resistance)[26]. Despite the absence of the receptor-cUMP/cCMP complex structure, the apo structure of cUMP synthase *Bc*PycC from *Burkholderia cepacia* showed that *Bc*PycC has a bulky side-chain of Tyr and Arg residues extending into the substrate-binding pocket, which sterically occludes the binding of larger purine nucleotides[26]. Thus, cyclic mono-pyrimidine and cyclic di-pyrimidine synthases in

other side. With uracil as the base, the pyrimidine ring remains virtually identical (Fig. 7B). However, similar flipping over of the cytosine base would result in the positional exchange of the 2-carbonyl and 4-amino groups, possibly causing some less favored interactions with the enzyme. Subtle changes in the substrate and intermediate binding modes may lead to significant differences in the catalyzed reactions.

Up to date, most of the identified bacterial CDNs contain 3'3'-phosphodiester linkage. One of the exceptions is 3',2'-cGAMP synthesized by CD-NTase in clade G from *Asticcacaulis* sp.[14], which could activate the downstream Cap5 effector for its specific defense against phage infections. Another case is the 2'3'3'-cyclic AMP-AMP-

bacteria utilize similar molecular mechanisms for pyrimidine nucleotide selection.

## Methods

### Protein expression and purification

Full-length gene fragments encoding *Cl*CdnE from *Cecembia lonarensis* (GenBank ID: EKB47661.1), *Ef*CdnE from *Enterococcus faecalis* (GenBank ID: EOK48090.1) and *Lp*CdnE from *Legionella pneumophila* (NCBI accession: WP_042646516.1) with *Escherichia coli* codon optimization were chemically synthesized and were each sub-cloned into pET21 vector by Blossom Biotechnologies, Inc. (Supplementary Table 7) to generate C-terminal His$_6$-tagged proteins. The constructed plasmids were transformed separately into *E. coli* BL21 (DE3), which was grown overnight in Luria Bertani (LB) broth at 37 °C. Overexpression of the target protein was induced by 0.5 mM IPTG when the OD$_{600}$ reached 0.6-0.8 and the bacteria were further incubated for 20 hours at 16 °C. The cells were harvested by centrifugation at 4 °C and 4500 x g for 30 min. The pellets were re-suspended in a lysis buffer containing 50 mM Tris-HCl pH 8.0, 500 mM NaCl, 10% glycerol, 1 mM tris(2-carboxyethyl)phosphine (TCEP), 1 mM phenylmethylsulfonyl fluoride (PMSF), and 5 mM imidazole. The cells were then lysed by sonication on ice. After centrifugation at 28000 x g, 4 °C, for 30 min, the clarified supernatant was loaded onto a 5 ml HisTrap FF column (Cytiva), followed by wash and elution with the lysis buffer that contained 10-200 mM imidazole gradient using ÄKTAprime plus (Cytiva). Based on subsequent analyses by 12% SDS-PAGE, fractions containing the target protein were pooled and further purified by gel-filtration using HiLoad 16/600 Superdex 200 pg column (Cytiva) equilibrated with a buffer T of 50 mM Tris-HCl pH 8.0, 200 mM NaCl, 5% glycerol and 1 mM TCEP.

### Crystallization and X-ray data collection

The sitting-drop vapor-diffusion method was employed in crystallizing the proteins and their nucleotide complexes using commercial kits from Hampton Research, Molecular Dimensions, and Emerald Biostructures. Each drop was set up by mixing 1 μl protein sample at a concentration of 12 mg/ml with 1 μl reservoir solution and equilibrated against 200 μl of the reservoir at 4 °C or 20 °C. The crystals of ligand-free form of *Cl*CdnE were grown by using a reservoir solution of 0.1 M MgCl$_2$, 0.1 M 2',2',2'-nitrilotriacetic acid (ADA) pH 6.5, 12% w/v polyethylene glycol (PEG) 6000. Complex crystals of *Cl*CdnE with UTP were obtained by including 5 mM UTP and 10 mM MgCl$_2$ in the protein solution. Ligand-free *Ef*CdnE was crystallized by using a reservoir solution containing 0.2 M sodium tartrate pH 7.3, 20% w/v PEG 3350. Both the complex crystals of *Ef*CdnE with the intermediate pppU[3'−5']pU and pppU[2'−5']p, were obtained by incubating the protein with 5 mM UTP and 10 mM MgCl$_2$ prior to setting up the drops.

Because *Lp*CdnE was prone to precipitate, crystals were not obtained until the concentration of protein sample was reduced to 2.4 mg/ml. Substrate-free *Lp*CdnE was crystallized by using a reservoir solution of 0.2 M MgCl$_2$, 0.1 M Tris pH 8.0, 5% w/v PEG 8000. The protein was co-crystallized with UTP by including 5 mM UTP and 10 mM MgCl$_2$, using a reservoir solution of 0.2 M ammonium acetate, 0.1 M sodium citrate pH 6.0, 9% w/v PEG 4000. The complex crystals of *Lp*CdnE with the non-hydrolysable UTP analogue UMPnPP, in which the O atom connecting the α- and β-phosphate is replaced by an imido (NH) group, were obtained by including 5 mM UMPnPP and 10 mM MgCl$_2$ in the protein solution and using 0.2 M lithium sulfate, 0.1 M Tris pH 8.5, 30% w/v PEG 3350 as the reservoir.

Before flash vitrification in liquid nitrogen, each protein crystal was washed briefly in the reservoir solution that contained 15−25% glycerol as a cryoprotectant. All of the X-ray diffraction data in this study were collected at the National Synchrotron Radiation

Research Center (NSRRC), Taiwan, and processed by using HKL2000_v722. The cubic *Cl*CdnE crystals diffracted X-rays to 2.2-2.6 Å resolution. They have space group I23 or I2$_1$3 and a single protein molecule in an asymmetric unit. The monoclinic crystals of *Ef*CdnE diffracted X-rays to 1.6−1.75 Å. They have space group P2$_1$ and an asymmetric unit also contains one protein molecule. The native *Lp*CdnE crystal diffracted X-rays to 2.5 Å, with trigonal space group P3$_1$ or P3$_2$ and one protein molecule in an asymmetric unit. The UTP and UMPnPP complex crystals of *Lp*CdnE with space groups P2$_1$ and P2$_1$2$_1$2 diffracted X-rays to 1.95 and 2.2 Å resolution. Both have two protein molecules in an asymmetric unit. Some data-collection statistics from the *Cl*CdnE, *Ef*CdnE and *Lp*CdnE crystals can be found in Supplementary Table 1−3.

### Structure determination and refinement

The protein sequence of *Cl*CdnE shows 36% identity to *Em*CdnE. Consequently, the *Em*CdnE model from PDB 6E0M was employed in solving the native *Cl*CdnE structure by molecular replacement (MR), using the program CNS[27]. The space group was confirmed as I23. Subsequent auto-building by PHENIX 1.19[28] and iterative manual rebuilding and adjustment of the model using Coot 0.9.6[29] yielded a refined model of native *Cl*CdnE, containing a continuous polypeptide chain from residues 2 to 298, the last three (L-E-H) corresponding to the His-tag, and 295 water molecules. The R and R$_{free}$ values were further improved by using Translation-Libration-Screw (TLS) tensors and the final values were 0.157 and 0.215, respectively, at a 2.2-Å resolution. Almost all (99.3%) residues were in the favored regions of the Ramachandran plot, and there were no outliers (Supplementary Table 1). The *Cl*CdnE-UTP complex crystal is isomorphous to the native. The *Fo-Fc* map showed two bound nucleotides in the active site, which were included in the model along with an Mg ion. The model was subsequently adjusted and refined by using COOT and PHENIX. Some statistics of the refined *Cl*CdnE structures can be found in Supplementary Table 1.

The native crystal structure of *Ef*CdnE was solved by MR, using CNS, in which the refined *Cl*CdnE structure served as the search model. The three-dimensional structure was iteratively refined and rebuilt using PHENIX and Coot and the TLS tensors were also applied. The refined model gave R and R$_{free}$ values of 0.168 and 0.198, respectively (Supplementary Table 2). The entire protein chain, including six histidine residues at the C-terminus, was visible. However, the connecting region between helices α1 and α2 has weak electron density. The *Ef*CdnE-pppU[3'−5']pU and *Ef*CdnE-pppU[2'−5']p complex crystals are also isomorphous to the native, and the bound intermediates, visible in the initial *Fo-Fc* and 2*Fo-Fc* maps, were placed into the active site. Some statistics of the *Ef*CdnE models after further refinement by using Coot and PHENIX can be found in Supplementary Table 2.

The native crystal structure of *Lp*CdnE was solved also by MR, but using MOLREP 11.7.03[30] and a search model from SWISS-MODEL[31] with *Cl*CdnE as the template. The space group was found to be P3$_1$, and the adjustment and refinement of the model were performed using Coot and REFMAC version 5.8.0267[32]. The final R and R$_{free}$ values were 0.191 and 0.223, respectively (Supplementary Table 3). The two complex structures of *Lp*CdnE were also solved by using MOLREP, with the native structure as the search model. Two protein molecules were successfully located in each unit cell. The bound ligands were placed into each active site and adjusted manually by using Coot. The complex structures were then refined by REFMAC, in which non-crystallographic symmetry (NCS) restraints and TLS tensors were used. Further refinements were carried out by using Coot and PHENIX. Some statistics of the refined *Lp*CdnE models can be found in Supplementary Table 3. All the figures containing three-dimensional protein structures were depicted using PyMOL 2.3.5[33].

## Affinity measurement

*Ef*CdnE was chosen as the sample to evaluate the affinity of this class of enzyme to different nucleotides by isothermal titration calorimetry (ITC), using nano-ITC (TA Instruments). *Ef*CdnE was dialyzed overnight against an assay buffer containing 20 mM Tris pH 8.0, 150 mM NaCl. The nucleotides to be tested (UTP, CTP, ATP and GTP, and their analogs) were each dissolved in the same buffer. Titration was carried out by 20 sequential injections of the nucleotide at a concentration of 250 μM at 3-min intervals to 50 μM *Ef*CdnE, with a stirring speed of 300 rpm. The heat of dilution resulting from injecting the nucleotide into the assay buffer was subtracted. The final data were analyzed and curve-fitted with sequential two-site mode by using NanoAnalyze v3.12.5 (TA Instruments).

## LC-MS/MS analysis

To analyze the intermediate and cyclic dinucleotide products synthesized by CdnE, reaction mixtures containing 1 mM $Mg^{2+}$, 0.5 mM indicated nucleotide substrates and 1 μM wild-type or mutant CdnE were prepared and overnight incubated. Reactions were heat inactivated at 90 °C for 10 mins, followed by centrifugation at 13000 x g, 4 °C for 30 min. After removal of the pellet, 5 μl of the sample solution was directly injected into the LC-MS system for analysis.

The LC-MS/MS experiments were performed as previously described[34] with some modifications. A UHPLC system (Ultimate 3000; Dionex, Germany) equipped with a XBridge BEH amide column (2.5um 2.1 × 150 mm; Waters) was coupled to a hybrid Q-TOF mass spectrometer (maXis impact, Bruker Daltonics, Bremen, Germany) with an orthogonal electrospray ionization (ESI) source. LC-MS data acquisition system was controlled by HyStar software (version 3.2, Bruker Daltonics). Mobile phase A was $H_2O$ that contained 10 mM ammonium bicarbonate. Mobile phase B was pure acetonitrile. The flow rate was 0.25 mL/min and an elution gradient was applied, starting with 90% B for 0.5 min, then down to 30% B in the next 6.5 min, followed by holding at 30% B for 1 min. It was then returned to the starting conditions by re-equilibration of the column for 3.2 min with 90% B prior to the next injection. The total run time was 11.2 min. The ESI source was operated in the negative mode (−3000 V). The endplate offset was 500 V. The nebulizer gas flow was 1 bar and drying gas flow was 8 L/min. The drying temperature was set at 200 °C. Funnel 1 radiofrequency (RF) and Funnel 2 RF were both 400 Vpp. The hexapole RF was 200 Vpp and the low mass cutoff of quadrupole was 100 m/z. To acquire MS/MS spectra of target ions, parallel reaction monitoring (PRM) was applied to acquired fragmented ions with the detection range of m/z 50 ~ 1000 at 2 Hz. The PRM collision energy was ramped from 32-48 eV for cUU, cUC, cCC, pppUpU, pppCpC and pppCpU/pppUpC. The precursor ion/fragment ion (observed retention time) of 611.04/305.01, 609.07/304.03 and 610.06/305.01 were extracted for the LC-MS peaks of cUU (RT: 6.4 min), cCC (RT: 6.1 min) and cUC (RT: 6.2 min), respectively. The isolation window for targets was 5 Da. The compound identification by the LC-MS/MS approach may be affected by the MS/MS spectra consisting of fragmented-ions from the co-eluting peaks with m/z difference smaller than the isolation window. All LC-MS raw data were processed by DataAnalysis (version 4.4, Bruker Daltonics) to obtain LC-MS spectra.

## Bioinformatic analysis

Multiple sequence alignment was performed using Clustal Omega[35]. Phylogenetic tree shown in Fig. 3C was generated using W-IQ-TREE[36]. To generate sequence logo shown in Figs. 5D and 8B–E, 100 CdnE protein sequences, 288 CdnA01 protein sequences, 31 CdnB04 protein sequences, 265 CdnC01 protein sequences, and 115 CdnG10 protein sequences were aligned using Clustal Omega, which then subjected to WebLogo (version 2.8.2)[37].

## Reporting summary

Further information on research design is available in the Nature Portfolio Reporting Summary linked to this article.

## Data availability

Data are available within the article and supplementary information. The coordinates and structure factors of *Cl*CdnE, *Ef*CdnE, *Lp*CdnE, *Cl*CdnE-UTP complex, *Ef*CdnE-pppU[3′−5′]pU complex, *Ef*CdnE-pppU[2′−5′]p complex, *Lp*CdnE-UTP complex and *Lp*CdnE-UMPnPP complex generated in this study have been deposited in the Protein Data Bank under accession codes 7X4A, 7X4C, 7X4F, 7X4G, 7X4P, 8HYK, 7X4Q and 7X4T. The protein structures used for analysis in this study are available in the Protein Data Bank under accession codes 4XJ3, 6E0K, 6E0L, 6E0M, 6E0N, 6E0O, 6P80, 7LJN, and 7LJO. The LC-MS dataset supporting the conclusions of this article is available from MassIVE (https://massive.ucsd.edu) under accession code MSV000092570.

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

## Acknowledgements
The authors thank National Synchrotron Radiation Research Center for beam time allocations and data collection supports. We also thank the Academia Sinica Protein Clinic, funded by Academia Sinica Core Facility and Innovative Instrument Project [AS-CFII-111-206] for the X-ray crystallographic service. This work was supported by the Ministry of Science and Technology of Taiwan [109-2311-B241-001 and 111-2311-B-039-001-MY3] (to Y.C.).

## Author contributions
C.S.Y. and T.P.K. designed and carried out the experiments and acquired and analyzed the data. C.J.C., M.H.H. and Y.C.W. carried out the experiments. Y.C. interpreted the data, wrote the manuscript, and supervised the entire project.

## Competing interests
The authors declare no competing interests.
