## [Peer Review File · Nature Communications]

Crystal structure and functional implications of cyclic di-pyrimidine-synthesizing cGAS/DncV-like nucleotidyltransferasesPoint-by-Point Response to the Reviewers' Comments

(Manuscript# NCOMMS-23-06658)

REVIEWER COMMENTS

Reviewer #1 (Remarks to the Author):

Yang and Ko et al present extensive structural and biochemical analysis demonstrating a mechanism for how bacterial CD-NTase enzymes in CBASS anti-phage defense can synthesize isomers of the nucleotide second messenger c-di-UMP. A highlight of the manuscript is the exceptionally thorough structural biology experiments including 8 different crystal structures of CD-NTase Clade E (CdnE) enzymes from three bacterial CBASS operons in apo and substrate-bound states. These CD-NTase enzymes are characterized biochemically for nucleotide substrate binding and cyclic dinucleotide synthesis. Overall, the experiments are well presented, and the new results are exciting for the field related to understanding of the biochemical basis of signaling specificity in anti-phage defense. Limitations of the manuscript relate to proposed CD-NTase cyclic dinucleotide products and lack of context of assessing potential nucleotide signals within downstream CBASS anti-phage defense. Otherwise, I only have minor control experiments and comments to help improve the manuscript for a general audience.

1) The authors propose 3'3'-c-di-UMP, 3'3'-c-di-CMP, and 2'3'-c-di-UMP as the major products of CBASS CdnE enzymes, but the identity of these nucleotide signals is only partially demonstrated with the current data. Three related points:

1A: The authors' conclusion of major nucleotide products for individual CdnE enzymes would be strengthened by including data from reactions where CD-NTase enzymes are provided all four common NTPs (A, G, C, U) in equal abundance.

1B: In several places throughout the text, the authors reference 3'3'-c-di-CMP produced by CdnE (lines 126, 145, 150, 352). Figures 2C and 2D show no evidence of c-di-CMP formation by LC/MS.

1C: The authors' conclusion that 2'3'-c-di-UMP is a product of CD-NTase enzymes is logical from the existing data, but it would be significantly more compelling if they could compare 2'3'-c-di-UMP against a chemical standard using MS/MS or NMR. In absence of these data, the authors should at least test susceptibility of 2'3'-c-di-UMP to Nuclease P1 cleavage.

2) An important unanswered question from this manuscript is if the proposed ClCdnE, EfCdnE, and LpCdnE nucleotide products are functional CBASS nucleotide signals. All known CD-NTase enzymes function by signaling through a downstream Cap (CD-NTase associated protein) effector that binds to the product nucleotide signal and causes cell death. It would be helpful if the authors include a schematic of the full CBASS operons for ClCdnE, EfCdnE, and LpCdnE in the main text. Are these CD-NTase enzymes encoded in homologous operons? Which Cap effector protein type is predicted to control phage defense?

Most importantly, the authors should consider experimentally testing if 3'3'-c-di-UMP and 2'3'-c-di-UMP function as signals to control downstream Cap activation. These experiments are not essential for publication, but they would significantly increase the impact of the authors' study. If the nucleotide signals cannot be verified to function in downstream signaling the authors should edit the text of the paper to make this point clear especially in terms of the potential discovery of 2'3'-c-di-UMP as a new second messenger in CBASS defense.

3) An interesting class of Clade E CD-NTases are FsCdnE and CgCdnE from STING-containing CBASS operons that function through synthesis of the nucleotide signal 3'3'-c-di-GMP (Morehouse et al. 2020 PMID 32877915). Can the authors compare their new CdnE structures against the previously reported FsCdnE and CgCdnE structures? What residues are encoded in the putative CD-NTase lid site 3 positions and is there a structural explanation for synthesis of 3'3'-c-di-GMP vs. 3'3'-c-di-UMP?

Minor Points

4) The authors conclude that lid 2 residues play no role in determining nucleotide specificity of these CD-NTases (line 248). While there is strong structural support for this conclusion based on the contacts made/not made by these residues, it is possible that lid 2 plays a role in nucleotide specificity during catalysis or through some other mechanism. This conclusion should be amended in the text or supported by biochemical experiments showing that mutation of lid 2 residues does not impact nucleotide selectivity.

5) The evidence suggesting that the lid 3 (R/Q)xW motif is critical to donor substrate specificity is strong. The authors confirm this finding by testing alanine mutations made to the arginine and tryptophan residues in EfCdnE which restored purine binding. It would be helpful to the argument if the same tests were performed with LpCdnE, mutating the Q residue to alanine.

6) The authors experiments with EfCdnE mutations R279 and W281 that reprogram

enzyme specificity are very interesting. Have the authors tested the activity of these enzymes with UTP alone, ATP alone, or GTP alone to determine if it's possible for the mutated active sites to synthesize cyclic di-purine or di-pyrimidine molecules?

7) The X-ray crystallography data statistics for the EfCdnE structures in Table S2 are atypical. The data appear significantly stronger than the current processing limit and further refinement and model building may yield more insight into the CD-NTase active site of these structures. Can the authors explain why the data were processed with such a high I/o and Rpim cutoff?

8) Please describe how the weblogo alignments were made in the methods section, how many CdnE enzymes were used for these alignments?

9) Text changes

a. Line 165 should reference Supplementary Figure S4, not S1.

b. Line 349 italicize Ef to correct EfCdnE

c. Line 358 says “the observation that cUC is more likely produced via the intermediate pppUpC rather than pppCpU is also consistent with” the authors’ explanation for the lack of cCC generation. However in line 148–149 they state that they cannot distinguish these two intermediates. It is not clear why the authors conclude that cUC is more likely produced via pppUpC rather than pppCpU.

d. Line 376: “We found that lid 2 of VcDncV (Val264) was sufficient to maintain the van der Waals interaction with the donor nucleotide but could not control its specificity. There is no direct evidence in the paper that Val264 does not control nucleotide specificity. The authors do not test VcDncV or mutate this residue to support this conclusion. Wordage should be changed to indicate that these conclusions about the function of Val264 is speculative and based on structure.

I hope the authors will find my comments useful, thank you for the opportunity to read this exciting manuscript.

Reviewer #2 (Remarks to the Author):

The article from Yang CS et al., describes a complete characterization of the cyclic di-pyrimidine-synthases: LpCdnE, ClCdnE and EfCdnE. The scope of the work is clear, and the manuscript is well written. The crystal structures as well as the binding capacities are well described and presented.

In my opinion the article is suitable for Nature communications.

I can only see few flaws and I have only few minor questions focused on the LC-MS identification and quantification methodologies used in this work.

I would recommend the authors to state clearly in the method the exact transitions and RT for all compound that you identified. This information is missing, and it will help the readers to follow better your experiment design.

I would also suggest to provide, if known, a reference for the LC-MS method even if partially or entirely inspired or changed.

Is there a reason why to use an isolation window of 5 Da?

I am happy to see that the authors reported the full extracted ion chromatogram (EIC) of the reaction intermediated in Figure 2. The fact that there are some signals before the actual peaks could be attribute to noise and it shows that the abundancy of these species is low. This information is also observable by the intensities and the number of the fragments in the MS2. I suggested to incorporate these comments in the text in order that the readers are not surprised by these figures.

In line with my previous comment, the MS2 of the compound seems to have lower number of fragments in comparison to previous published data. Did the authors try lower or higher collision energies? The authors don't need to repeat the experiments they can state or simply explain what the case is here and how they can relate to published papers.

The separation and elution of cDNs species is complex mainly because they have similar structure and they often coelute. I also have experience with these molecules and in some cases, it is possible to separate them in other not. In Figure S9 all compound eluting at the same time. I can imagine that this can affect the identification and prevent obtain a rich MS2 while isolating four species at the same time. I suggest the author to add few lines on the limitation of the method.

In Figures S2 and S9 two peaks are observed in the extracted chromatogram. Are they both the compound of interest? Additionally, they are not present in Figure 6. Please provide an explanation.

Minor comments:

Please increase the resolution of figure S3 and obtain a better visualization for Figure 2.

Author Rebuttals

Reviewer #1 (Remarks to the Author):

Yang and Ko et al present extensive structural and biochemical analysis demonstrating a mechanism for how bacterial CD-NTase enzymes in CBASS anti-phage defense can synthesize isomers of the nucleotide second messenger c-di-UMP. A highlight of the manuscript is the exceptionally thorough structural biology experiments including 8 different crystal structures of CD-NTase Clade E (CdnE) enzymes from three bacterial CBASS operons in apo and substrate-bound states. These CD-NTase enzymes are characterized biochemically for nucleotide substrate binding and cyclic dinucleotide synthesis. Overall, the experiments are well presented, and the new results are exciting for the field related to understanding of the biochemical basis of signaling specificity in anti-phage defense. Limitations of the manuscript relate to proposed CD-NTase cyclic dinucleotide products and lack of context of assessing potential nucleotide signals within downstream CBASS anti-phage defense. Otherwise, I only have minor control experiments and comments to help improve the manuscript for a general audience.

1) The authors propose 3'3'-c-di-UMP, 3'3'-c-di-CMP, and 2'3'-c-di-UMP as the major products of CBASS CdnE enzymes, but the identity of these nucleotide signals is only partially demonstrated with the current data. Three related points:

1A: The authors' conclusion of major nucleotide products for individual CdnE enzymes would be strengthened by including data from reactions where CD-NTase enzymes are provided all four common NTPs (A, G, C, U) in equal abundance.

Response: Thanks for reviewer's suggestion. We have first proved that *EfCdnE* did not bind purine nucleotide substrates, ATP and GTP, but instead recognizes UTP and CTP with submicromolar affinity via ITC experiments. Therefore, we excluded ATP and GTP for further product analysis of *EfCdnE* and found that by providing both UTP and CTP as substrates in equal abundance, *EfCdnE* produced c-di-UMP and c-UMP-CMP as products (Fig. 2F-2G). As the noises before the actual peaks are obvious (Fig. 2F-2G), simultaneous addition of all four NTPs, including ATP, GTP, CTP, and UTP, will further increase the noises and thus complicate the MS/MS spectra, resulting in difficulty in differentiating the fragmented-ions.

1B: In several places throughout the text, the authors reference 3'3'-c-di-CMP produced by CdnE (lines 126, 145, 150, 352). Figures 2C and 2D show no evidence of c-di-CMP formation by LC/MS.

Response: We greatly appreciate the reviewer's comment. In LC-MS/MS experiments, we have detected a large sharp peak corresponding to pppCpC by incubating *EfCdnE* with CTP alone or both CTP and UTP in more than three independent experiments. However, the amount of c-di-CMP products were approaching the limits of the instrument used. As described in line 351-357 in the discussion section, the rearrangement and re-binding of the intermediate pppCpC probably cause significant clashes owing to the larger 4-amino group of cytosine base in contrast to the carbonyl group of uracil base. It is thus suggested that the energy barrier for formation of c-di-CMP product from intermediate pppCpC is too high to overcome, leading to the extremely low yield of c-di-CMP. For the preciseness and rigorousness, we decided to remove all the EIC of c-di-CMP after careful consideration and discussion with mass spectrometry experts. As a result, new Figures 2D and 2E only display peaks corresponding to pppCpC. The description about c-di-CMP production in lines 127, 145-146 and 350-351 were modified in the revised manuscript. The sentence "...and cCC products through pppCpC as an intermediate, although in low amounts" in original line 150 has been removed.

IC: The authors' conclusion that 2'3'-c-di-UMP is a product of CD-NTase enzymes is logical from the existing data, but it would be significantly more compelling if they could compare 2'3'-c-di-UMP against a chemical standard using MS/MS or NMR. In absence of these data, the authors should at least test susceptibility of 2'3'-c-di-UMP to Nuclease P1 cleavage.

Response: We greatly thanks for the reviewer's suggestion. Characterization of cyclic GMP-AMP (cGA) using chemical standards 3'3'-cGA and 2'3'-cGA for comparison in MS/MS analysis has been widely used. However, in our case, the chemical standards 2'3'-c-di-UMP required for comparison are currently unpurchasable. We have inquired whether chemical vendors could synthesize specific isomer 2'3'-c-di-UMP but could not get positive responses. Unfortunately, I don't have the expensive NMR equipment and well-trained operators in my lab to determine the chemical structure of the final product of CdnE enzymes. The alternative way to access the phosphodiester linkage of cyclic dinucleotides is to treat the cyclic dinucleotides generated by CdnE using P1 nuclease or snake venom phosphodiesterase (SVPD) for comparison. P1 nuclease specifically cleave 3',5'-phosphodiester linkage but not 2',5'-phosphodiester linkage, whereas SVPD cleaves both 3',5'- and 2',5'-phosphodiester linkages. The most sensitive method is utilizing radiolabeled nucleotide substrates, such as α -³²P-ATP or α -³²P-GTP, to produce radiolabeled CDN products followed by digestion via P1 nuclease or SVPD. The final product 3'3'-cGA will be completely digested to single nucleotides AMP and GMP in contrast to the partial digestion of product 2'3'-cGA, which contains one 2',5'-phosphodiester linkage that resist P1 nuclease digestion. However, experiments with

radioisotopes are unavailable in our institution due to regulatory restrictions. In addition, to accurately define the composition of phosphodiester linkages of c-di-UMP, we also require chemical standards, such as pppU[2'-5']pU, to identify the digested products, which are currently unpurchasable. Therefore, it is regrettable that the key finding regarding production of specific isomer of c-di-UMP by CdnE enzyme is limited to structural observations. We will try collaborating with other labs in the future to further investigate whether CdnE enzymes produce 2'3'-c-di-UMP and whether 2'3'-c-di-UMP activates the downstream Cap proteins.

2) An important unanswered question from this manuscript is if the proposed *CICdnE*, *EfCdnE*, and *LpCdnE* nucleotide products are functional CBASS nucleotide signals. All known CD-NTase enzymes function by signaling through a downstream Cap (CD-NTase associated protein) effector that binds to the product nucleotide signal and causes cell death. It would be helpful if the authors include a schematic of the full CBASS operons for *CICdnE*, *EfCdnE*, and *LpCdnE* in the main text. Are these CD-NTase enzymes encoded in homologous operons? Which Cap effector protein type is predicted to control phage defense?

Response: We greatly thanks for the reviewer's suggestion. A schematic of the full CBASS operons for *CICdnE*, *EfCdnE*, and *LpCdnE* has been added to Fig. 2A (see below). All these three CBASS operons encode one CdnE enzyme and one transmembrane (TM) effector and belong to type I-B based on previous classification (Millman et al. *Nature microbiology* 2020, PMID: 32839535). Based on sequence homology, the downstream TM effectors of *CICdnE*, *EfCdnE*, and *LpCdnE*, are further classified as Cap15, which contain two transmembrane segments and cause inner membrane disruption to confer phage resistance. The following description "...,which are encoded in type I CBASS operons with a transmembrane (TM) effector, named Cap15" and "A schematic of the full CBASS operons for *CICdnE*, *EfCdnE*, and *LpCdnE*. The effector Cap15 contains two transmembrane segments and a β -barrel and has been shown to cause inner membrane disruption upon ligand binding and oligomerization" have been added in line 118-119 and 711-713 of the revised manuscript, respectively.

Fig. 2A. A schematic of the full CBASS operons for *CICdnE*, *EfCdnE*, and *LpCdnE*. The effector Cap15 contains two transmembrane segments and a β -barrel and has been shown to cause inner membrane disruption upon ligand binding and oligomerization.

Most importantly, the authors should consider experimentally testing if 3'3'-c-di-UMP and 2'3'-c-di-UMP function as signals to control downstream Cap activation. These experiments are not essential for publication, but they would significantly increase the impact of the authors' study. If the nucleotide signals cannot be verified to function in downstream signaling the authors should edit the text of the paper to make this point clear especially in terms of the potential discovery of 2'3'-c-di-UMP as a new second messenger in CBASS defense.

Response: We greatly thanks for the reviewer's suggestion. Owing to the unpurchasable c-di-UMP isomer, 2'3'-c-di-UMP, we could not perform *in vitro* experiments to investigate whether specific isomers of c-di-UMP activates the downstream Cap effector of *EfCdnE*. To clarify this issue, the sentence "... which will lead to the production of a novel second messenger, 2'3'-cUU" was modified to "...which will lead to the production of 2'3'-cUU" in line 323 of the revised manuscript and the sentence "However, whether specific isomer of cUU activates the downstream Cap15 effector remains unclear and need to be characterize in the future study" has been added to the discussion section in line 366-368 of the revised manuscript.

*3) An interesting class of Clade E CD-NTases are *FsCdnE* and *CgCdnE* from STING-containing CBASS operons that function through synthesis of the nucleotide signal 3'3'-c-di-GMP (Morehouse et al. 2020 PMID 32877915). Can the authors compare their new *CdnE* structures against the previously reported *FsCdnE* and *CgCdnE* structures? What residues are encoded in the putative CD-NTase lid site 3 positions and is there a structural explanation for synthesis of 3'3'-c-di-GMP vs. 3'3'-c-di-UMP?*

Response: Structural comparison of *FsCdnE* (PDB: 6WT8) and *CgCdnE* (PDB: 6WT9) with cyclic di-pyrimidine-synthesizing *CdnEs* in this study show similar overall architecture with r.m.s.d. ranging from 2.6 to 3.2 Å. As shown in Figure below, both *FsCdnE* and *CgCdnE* have two additional α -helices at the N-terminus compared with *CiCdnE*, *EfCdnE* and *LpCdnE* (A-F). Additionally, the loop region between β B' and α 3 of *CiCdnE*, *EfCdnE* and *LpCdnE* is replaced by two α -helices in *FsCdnE*, but invisible in *CgCdnE* (A-F). Analysis of their active sites revealed that in contrast to the conserved asparagine residue (N169 of *CiCdnE*) for uracil base selection, both *FsCdnE* and *CgCdnE* contains an aspartate residue, D233 and D226, respectively, at lid 1 position (G-I). Previous report (Morehouse et al., 2020, PMID: 32877915) demonstrated that mutating D233 to alanine reduced the enzyme activity and c-di-GMP production by *FsCdnE*, supporting its critical role for guanine base recognition. Furthermore, multiple sequence alignment of *FsCdnE* related homologs from STING-containing CBASS operons revealed a conserved DxxE motif (D326 and E329 of *FsCdnE*) at the C-terminal loop above the donor pocket (I). Despite lacking GTP-bound *FsCdnE* structure, either D326 or E329

could form bidentate hydrogen bonds with the guanine base, thus explaining the synthesis of c-di-GMP by *FsCdnE* related proteins. In conclusion, to achieve substrate specificity, cyclic di-GMP-synthesizing CdnE contains an aspartate residue as lid 1 and DxxE motif as lid 3, whereas cyclic di-pyrimidine-synthesizing CdnE utilize an asparagine residue as lid 1 and (R/Q)xW as lid 3.

Structural comparison of *FsCdnE* (PDB: 6WT8) and *CgCdnE* (PDB: 6WT9) with *C/CdnE*, *EfCdnE* and *LpCdnE*. (A-C) Superimposition of *FsCdnE* (green) with (A) *C/CdnE* (red), (B) *EfCdnE* (orange) and (C) *LpCdnE* (magenta). (D-F) Superimposition of *CgCdnE* (cyan) with (D)

CICdnE (red), (E) *EfCdnE* (orange) and (F) *LpCdnE* (magenta). The major structural differences between them are highlighted by black dashed circles. (G-H) Enlarged view of the nucleotide binding pockets of UTP-bound *CICdnE* superimposed with (G) *FsCdnE* and (H) *CgCdnE*. The Lid 1 residues and bound UTP analogs are shown in sticks. (I) Sequence alignment of *FsCdnE* related homologs from STING-containing CBASS operons. *MyCdnE* from *Myroides sp.* ZB35 (GenBank: APA92223) and *PcCdnE* from *Prevotella corporis* (GenBank: KXA32417) produce c-di-GMP that specifically activates their downstream STING effectors (Ko et al., 2022, PMID: 35013136). Lid 1, 2, and 3 are indicated by arrows.

Minor Points

4) *The authors conclude that lid 2 residues play no role in determining nucleotide specificity of these CD-NTases (line 248). While there is strong structural support for this conclusion based on the contacts made/not made by these residues, it is possible that lid 2 plays a role in nucleotide specificity during catalysis or through some other mechanism. This conclusion should be amended in the text or supported by biochemical experiments showing that mutation of lid 2 residues does not impact nucleotide selectivity.*

Response: Thanks for reviewer's suggestion. The original sentence "However, lid 2 of *LpCdnE* and *CICdnE* structures did not determine the donor nucleotide specificity" was removed and the description "suggesting that lid 2 may plays a role in nucleotide specificity during catalysis or through some other mechanism" has been added in line 249-250 of the revised manuscript.

5) *The evidence suggesting that the lid 3 (R/Q)xW motif is critical to donor substrate specificity is strong. The authors confirm this finding by testing alanine mutations made to the arginine and tryptophan residues in EfCdnE which restored purine binding. It would be helpful to the argument if the same tests were performed with LpCdnE, mutating the Q residue to alanine.*

Response: We appreciate the reviewer's suggestion. We have cloned, expressed, and purified the Q274A mutant of *LpCdnE* using the same procedure as wild-type *LpCdnE*. As shown in the figure below, *LpCdnE*^{Q274A} proteins were prepared by a two-step procedure using Ni-NTA affinity chromatography followed by size-exclusion chromatography (A-D). The mutant proteins were purified to monodisperse and high-purity (C, D). However, *LpCdnE*^{Q274A} proteins became extremely unstable and rapidly precipitated during ITC experiments (E).

Preparation of *LpCdnE*^{Q274A} proteins. (A) Ni-NTA affinity chromatography of *LpCdnE*^{Q274A}. (B) The 12 % SDS-PAGE of purified *LpCdnE*^{Q274A} in (A). Lane M: Marker. Lane FT: flow-through. The expected size of *LpCdnE*^{Q274A} protein is about 34 kDa. The fraction 27–38 was pooled, concentrated, and subjected to size-exclusion chromatography. (C) Size-exclusion chromatography of *LpCdnE*^{Q274A}. (D) The 12 % SDS-PAGE of purified *LpCdnE*^{Q274A} in (C). (E) *LpCdnE*^{Q274A} proteins precipitated rapidly in the assay buffer used for ITC experiments.

6) The authors experiments with *EfCdnE* mutations R279 and W281 that reprogram enzyme specificity are very interesting. Have the authors tested the activity of these enzymes with UTP alone, ATP alone, or GTP alone to determine if it's possible for the mutated active sites to synthesize cyclic di-purine or di-pyrimidine molecules?

Response: Thanks for reviewer's comment. The suggested experiments were performed at three independent replicates. The results were shown in the figure below. In contrast to wild-type *EfCdnE*, either R279A or W281A mutant of *EfCdnE* generated no c-di-UMP product after incubation with UTP substrates (A, F), supporting our finding that (R/Q)xW motif is critical for pyrimidine selection. Furthermore, no cyclic di-purine products were detected by including either ATP alone or GTP alone as substrates in the reaction mixture with *EfCdnE*^{R279A} or *EfCdnE*^{W281A} (B-E), supporting the important role of the conserved asparagine residue at lid 1 position dictating pyrimidine specificity.

Functional validation of the dual pyrimidine selection of *EfCdnE*. (A–C) Mass spectrum from a sample of *EfCdnE*^{R279A} incubated with UTP (A), ATP (B) or GTP (C). (D–F) Mass spectrum from a sample of *EfCdnE*^{W281A} incubated with ATP (D), GTP (E) or UTP (F). Extracted ion chromatograms (EIC) of fragment ions of m/z 609.07 and m/z 304.03, corresponding to cyclic di-UMP, EIC of fragment ions of m/z 657.09 and m/z 328.04, corresponding to cyclic di-AMP and EIC of fragment ions of m/z 689.08 and m/z 344.03, corresponding to cyclic di-GMP were shown as indicated.

7) The X-ray crystallography data statistics for the *EfCdnE* structures in Table S2 are atypical.

The data appear significantly stronger than the current processing limit and further refinement and model building may yield more insight into the CD-NTase active site of these structures. Can the authors explain why the data were processed with such a high I/o and Rpim cutoff?

Response: Thanks for reviewer's comment. To get X-ray crystallography data with high quality, we usually processed the data with a stringent cut-off (>95%) for completeness. In addition, due to the limits of allocated beamtime and the difficulty in reproducing the *EfCdnE* crystals with the same quality, we could not further improve the structures.

8) Please describe how the weblogo alignments were made in the methods section, how many CdnE enzymes were used for these alignments?

Response: Thanks for reviewer's suggestion. The details of creating weblogo alignments has been described in the method section. The following description "To generate sequence logo shown in Fig. 5D and Fig. 8B–8E, 100 CdnE protein sequences, 288 CdnA01 protein sequences, 31 CdnB04 protein sequences, 265 CdnC01 protein sequences, and 115 CdnG10 protein sequences were aligned using Clustal Omega, which then subjected to WebLogo (version 2.8.2)" was added in line 537-540 in the method section.

9) Text changes

a. Line 165 should reference Supplementary Figure S4, not S1.

Response: Thanks for reviewer's suggestion. This error has been corrected in line 164 of the revised manuscript.

*b. Line 349 italicize Ef to correct *EfCdnE**

Response: Thanks for reviewer's suggestion. This error has been corrected in line 348 of the revised manuscript.

c. Line 358 says "the observation that cUC is more likely produced via the intermediate pppUpC rather than pppCpU is also consistent with" the authors' explanation for the lack of cCC generation. However in line 148–149 they state that they cannot distinguish these two intermediates. It is not clear why the authors conclude that cUC is more likely produced via pppUpC rather than pppCpU.

Response: Thanks for reviewer's comment. In LC-MS analysis, the molecular weight of pppUpC and pppCpU is identical, so we could not distinguish these two intermediates. Nevertheless, structural comparison of *CiCdnE*-UTP and *EfCdnE*-pppU[3'-5']pU shown

in Fig. 7B demonstrated that the binding mode of the uracil base of acceptor nucleotide between them is nearly identical, whereas the recognition of donor nucleotide are different. The re-orientation and re-binding of pppUpU intermediate caused flipping of uracil base and the rotation of ribose of the donor moiety. This will result in significant difference upon pppCpU binding and subsequent cyclization due to the larger 4-amino group of cytosine base in contrast to the carbonyl group of uracil base. However, after extensive crystallization trials, we still could not obtain the complex crystal of *EfCdnE*-pppCpU or *EfCdnE*-pppUpC. Due to lack of solid evidence, we decided to remove the sentence “The observation that cUC is more likely produced via the intermediate pppUpC rather than pppCpU is also consistent with the above hypothesis” for preciseness and rigorousness.

d. Line 376: “We found that lid 2 of VcDncV (Val264) was sufficient to maintain the van der Waals interaction with the donor nucleotide but could not control its specificity. There is no direct evidence in the paper that Val264 does not control nucleotide specificity. The authors do not test VcDncV or mutate this residue to support this conclusion. Wordage should be changed to indicate that these conclusions about the function of Val264 is speculative and based on structure.

Response: Thanks for reviewer’s suggestion. The description that “We found that lid 2 of *VcDncV* (Val264) was sufficient to maintain the van der Waals interaction with the donor nucleotide but could not control its specificity” was changed to “Based on current available structures, lid 2 of *VcDncV* (Val264) was found to make van der Waals interaction with the donor nucleotide but does not form base-specific interactions” in line 373–375 of the revised manuscript.

I hope the authors will find my comments useful, thank you for the opportunity to read this exciting manuscript.

Response: Thanks for the reviewer’s invaluable comments and suggestions to make this revised manuscript better. The majority of the questions have been properly answered. Some important issues regarding the functional CBASS signal produced by *CdnE* enzymes that we really want to address but lack resources to conduct the recommended experiments. I sincerely hope that you can find the useful and informative answers that we provided above. Thank you again for your time and expertise for reviewing our manuscript.

Reviewer #2 (Remarks to the Author):

The article from Yang CS et al., describes a complete characterization of the cyclic di-pyrimidine-synthases: LpCdnE, ClCdnE and EfCdnE. The scope of the work is clear, and the manuscript is well written. The crystal structures as well as the binding capacities are well described and presented.

In my opinion the article is suitable for Nature communications.

I can only see few flaws and I have only few minor questions focused on the LC-MS identification and quantification methodologies used in this work.

I would recommend the authors to state clearly in the method the exact transitions and RT for all compound that you identified. This information is missing, and it will help the readers to follow better your experiment design.

Response: In this study, PRM scan mode was used. In PRM, a precursor ion of interest is selected and fragmented to produce a series of product ions, which are then detected in a wide range from m/z 50 to m/z 1000. To clarify this issue, the sentence of “The ESI source was operated in the negative mode (-3000 V) with the detection range of m/z 50~1000 at 2 Hz” was modified to “The ESI source was operated in the negative mode (-3000 V)” in line 523 of the revised manuscript. The sentence of “To acquire MS/MS spectra of target ions, parallel reaction monitoring (PRM) was applied” was changed to “To acquire MS/MS spectra of target ions, parallel reaction monitoring (PRM) was applied to acquired fragmented ions with the detection range of m/z 50~1000 at 2 Hz” in line 527–528 of the revised manuscript. One new sentence was added in the method section as follows: “The precursor ion/fragment ion (observed retention time) of 611.04/305.01, 609.07/304.03 and 610.06/305.01 were extracted for the LC-MS peaks of cUU (RT:6.4 min), cCC (RT:6.1 min) and cUC (RT:6.2 min), respectively” in line 529–531 of the revised manuscript.

I would also suggest to provide, if known, a reference for the LC-MS method even if partially or entirely inspired or changed.

Response: Thanks for reviewer’s suggestion. We have applied an amide LC column and a triple quadrupole MS using MRM scan mode for the analysis of metabolites in our previously-published paper (Chen et al. J. Proteome Res 2018, 17, 3997–4007). In this study, a similar LC-MS method we used by changing a triple quadrupole MS with MRM to a hybrid Q-TOF MS with PRM scan mode to analyze the products.

Therefore, the above-mentioned reference was added in the revised manuscript and the

sentence of “A UHPLC system (Ultimate 3000; Dionex, Germany) equipped with a XBridge BEH amide column (2.5um 2.1 × 150 mm; Waters) was coupled to a hybrid Q-TOF mass spectrometer...” was modified to “The LC-MS/MS experiments were performed as previously described³³ with some modifications. A UHPLC system (Ultimate 3000; Dionex, Germany) equipped with a XBridge BEH amide column (2.5um 2.1 × 150 mm; Waters) was coupled to a hybrid Q-TOF mass spectrometer...” in line 515-517 of the revised manuscript.

Is there a reason why to use an isolation window of 5 Da?

Response: In this study, a high resolution and accurate MS, Q-TOF, was used to obtain LC-MS data for reliable results. However, because the low duty cycle of Q-TOF limits the sensitivity of Q-TOF systems, most Q-TOF mass spectrometers used a larger isolation window of several m/z units to increase the ion transmission and leads to higher detection sensitivity. The isolation window of 5 Da is the recommended and default setting in our Q-TOF system.

I am happy to see that the authors reported the full extracted ion chromatogram (EIC) of the reaction intermediated in Figure 2. The fact that there are some signals before the actual peaks could be attribute to noise and it shows that the abundancy of these species is low. This information is also observable by the intensities and the number of the fragments in the MS2. I suggested to incorporate these comments in the text in order that the readers are not surprised by these figures.

Response: Thanks for reviewer’s suggestion. The comments “Some signals before the actual peaks corresponding to intermediate pppUpU, pppCpC, and pppUpC/pppCpU in (D–G) could be attribute to noise and it shows that the abundancy of these species is low” is added to the legend of Figure 2 in line 727–729 of the revised manuscript.

In line with my previous comment, the MS2 of the compound seems to have lower number of fragments in comparison to previous published data. Did the authors try lower or higher collision energies? The authors don’t need to repeat the experiments they can state or simply explain what the case is here and how they can relate to published papers.

Response: Thanks for the reviewer’s comment. Because we did not find the MS/MS spectrum of cUU, cUC, cCC, we purchased the chemical standards for comparison. Because the elution time, MS and MS/MS of the target compounds (cUU, cUC and cCC) matches with that information of the chemical standards, these target compounds can be successfully identified. In MS/MS function by using a Q-TOF system, high collision energy

can produce lower m/z fragmented ions but losing higher m/z fragmented ions and vice versa. To have more abundant fragmented ions, a ramping collision voltage from 32–48 eV was used in this study. To clarify this issue, the sentence of “The PRM collision energy was 35 for cUU, cUC, cCC, pppUpU, pppCpC and pppCpU/pppUpC” was modified to “The PRM collision energy was ramped from 32–48 eV for cUU, cUC, cCC, pppUpU, pppCpC and pppCpU/pppUpC” in line 528–529 of the revised manuscript.

The separation and elution of cDNs species is complex mainly because they have similar structure and they often coelute. I also have experience with these molecules and in some cases, it is possible to separate them in other not. In Figure S9 all compound eluting at the same time. I can imagine that this can affect the identification and prevent obtain a rich MS2 while isolating four species at the same time. I suggest the author to add few lines on the limitation of the method.

Response: As the reviewer’s suggestion, the description “The compound identification by the LC-MS/MS approach may be affected by the MS/MS spectra consisting of fragmented-ions from the co-eluting peaks with m/z difference smaller than the isolation window” was added in line 531–533 of the revised manuscript to describe the limitation of this method.

In Figures S2 and S9 two peaks are observed in the extracted chromatogram. Are they both the compound of interest? Additionally, they are not present in Figure 6. Please provide an explanation.

Response: After checking the two peaks eluted at 5.5 min and 6.1 min, we observed that they have the same fragmented ions. Interestingly, we also noticed that the broadened peak at 5.5 min disappeared once the standard solution was diluted beyond a 2-fold dilution. Based on these observations, we propose that the broadened peak at 5.5 min could be a dimer formed by non-covalent bonding under high concentration. This dimer may have been fragmented into a monomer in the ESI interface, leading to the disappearance of the broadened peak. To avoid readers confusion, Figure S2 and S9 were replaced with the new figures, in which diluted standard solution was analyzed to have a single LC-MS peak.

New Figure S2. LC-MS/MS analysis of the chemical standards, cUU, cCC and cUC.

(A) LC-MS/MS result of cyclic di-UMP (theoretical [M-H]⁻: m/z 611.0422), which is eluted at 6.1 min and has fragmented ions of m/z 305.0209, m/z 384.9865, and m/z 499.0204. (B) LC-MS/MS result of cyclic di-CMP (theoretical [M-H]⁻: m/z 609.0741), which is eluted at 6.2 min and has fragmented ions of m/z 211.0019, m/z 304.0367, m/z 384.0027 and m/z 498.0364. (C) LC-MS/MS result of cyclic CMP-UMP (theoretical [M-H]⁻: m/z 610.0582), which is eluted at 6.1 min and has fragmented ions of m/z 211.0007, m/z 305.0196, m/z 323.0301, m/z 383.9997, m/z 498.0327 and m/z 567.0545.

New Figure S9. Functional validation of the critical role of (R/Q)xW motif of *EfCdnE*.

(A-B) ITC analysis of the interaction between (A) non-hydrolysable AMPcPP and *EfCdnE*^{R279A} and (B) non-hydrolysable GMPCPP and *EfCdnE*^{W281A}. The determined dissociation constants (K_D , M) were indicated. (C) LC-MS/MS result of cyclic UMP-AMP (theoretical [M-H]⁻: m/z 634.077), which is eluted at 6.1 min and has fragmented ions of m/z 211.0023, m/z 305.0169, m/z 328.0435 and m/z 522.0353. (D) A peak of m/z 633.03 and its fragmented ions was found to be eluted at 6.1 min having the same elution time of cyclic UMP-AMP. (E) LC-MS/MS result of cyclic UMP-GMP (theoretical [M-H]⁻: m/z 650.0643), which is eluted at 6.1 min and has fragmented ions of m/z 305.0169, m/z 344.0384, m/z 384.9829 and m/z 499.0191. (F), a peak of m/z 649.0 and its fragmented ions was found to be eluted at 6.1 min having the same elution time of cyclic UMP-GMP.

Minor comments:

Please increase the resolution of figure S3 and obtain a better visualization for Figure 2.

Response: Thanks for the reviewer's suggestion. The resolution of both Figure 2 and Figure S3 has been increased and the chemical structure of each fragment ion derived from intermediate pppUpU and pppUpC or pppCpU has been re-depicted using ChemDraw software. Figure 2 and S3 have been replaced with new one, which were also displayed below for your convenience.

New Figure 2. LC-MS/MS analysis of the reaction intermediate and products synthesized by

EfCdnE

(A) A schematic of the full CBASS operons for *CiCdnE*, *EfCdnE*, and *LpCdnE*. The effector Cap15 contains two transmembrane segments and a β -barrel and has been shown to cause inner membrane disruption upon ligand binding and oligomerization²⁰. (B) LC-MS/MS analysis of the reaction mixture containing purified *EfCdnE*, UTP and Mg^{2+} ion after overnight incubation. Top panel, a peak of precursor ion (m/z 611.0342) and its fragmented ions (m/z 305.0126 and m/z 499.0078) was detected at 6.4 min, which correspond to cUU according to the matched LC retention time and MS and MS/MS spectra with the chemical standard. Bottom panel, a peak of precursor ion (m/z 788.9751) and its numerous fragmented ions (see below) was detected at 7.4 min, which correspond to pppUpU. (C) The enlarged view of the mass spectrum shown in bottom panel in (A). The fragmented ions (m/z 709.0086, m/z 690.9987, m/z 629.044, m/z 611.032, m/z 593.0233, m/z 544.9114, m/z 464.9441, m/z 384.9771, m/z 305.0148) and their corresponding chemical structures derived from pppU[3'-5']pU were shown. (D) Mass spectrum from a sample of *EfCdnE* incubated with CTP. Extracted ion chromatograms (EIC) of a fragment ion (m/z 787.01), corresponding to pppCpC. (E-G) Mass spectrum from a sample of *EfCdnE* incubated with CTP and UTP. (E) EIC of a fragment ion (m/z 787.01), corresponding to pppCpC. (F) Top panel, EIC of a fragment ion (m/z 788.98), corresponding to pppUpU. Bottom panel, EIC of fragment ions of m/z 611.04 and m/z 305.01, corresponding to cUU. (G) Top panel, EIC of a fragment ion (m/z 788.00), corresponding to pppUpC or pppCpU. Bottom panel, EIC of fragment ions of m/z 610.05 and m/z 305.01, corresponding to cUC. Some signals before the actual peaks corresponding to intermediate pppUpU, pppCpC, and pppUpC/pppCpU in (D–G) could be attribute to noise and it shows that the abundancy of these species is low.

A

B

New Figure S3. LC-MS/MS analysis of the intermediates generated by *Ef*CdnE using both CTP and UTP as substrates.

The EIC of a fragment ion (m/z 788.00), corresponding to (A) pppUpC or (B) pppCpU. The chemical structures corresponding to each fragment ion derived from (A) pppUpC or (B) pppCpU are shown.

Point-by-Point Response to the Reviewers' Comments

(Manuscript# NCOMMS-23-06658A)

REVIEWER COMMENTS

Reviewer #1 (Remarks to the Author):

The authors have addressed most of my comments with new analysis and changes to the text. I recommend publication, and congratulate the authors on a very interesting study.

Response: We greatly appreciate this reviewer for the invaluable and insightful comments and suggestions to make our manuscript better.